# A bacterial network of T3SS effectors counteracts host pro-inflammatory responses and cell death to promote infection

Hui Wen Yeap [1,2], Ghin Ray Goh[1,2], Safwah Nasuha Rosli[1,2], Hai Shin Pung[1,2], Cristina Giogha[3,4], Vik Ven Eng[3,5], Jaclyn S Pearson [3,5,6], Elizabeth L Hartland[3,4,5] & Kaiwen W Chen [1,2 ✉]

## Abstract

Innate immune signalling and cell death pathways are highly interconnected processes involving receptor-interacting protein kinases (RIPKs) as mediators of potent anti-microbial responses. However, these processes are often antagonised by bacterial type III secretion system (T3SS) effectors, and the cellular mechanisms by which the host retaliates are not completely understood. Here, we demonstrate that during *Citrobacter rodentium* infection, murine macrophages and colonic epithelial cells exhibit RIPK1 kinase-dependent caspase-8 activation to counteract NleE effector-mediated suppression of pro-inflammatory signalling. While *C. rodentium* injects into the host cells a second effector, NleB, to block caspase-8 signalling, macrophages respond by triggering RIPK3-mediated necroptosis, whereupon a third T3SS effector, EspL, acts to inactivate necroptosis. We further show that NleB and EspL collaborate to suppress caspase-8 and NLRP3 inflammasome activation in macrophages. Our findings suggest that *C. rodentium* has evolved to express a complex network of effectors as an adaptation to the importance of cell death for anti-bacterial defence in the host-pathogen arms race.

Keywords Caspase-8; Necroptosis; Apoptosis; Pyroptosis; Infection
Subject Categories Autophagy & Cell Death; Immunology; Microbiology; Virology & Host Pathogen Interaction

## Introduction

Receptor-interacting serine/threonine-protein kinase 1 (RIPK1) is a pleiotropic mediator of cell death and inflammation that plays a key role during tumour necrosis factor receptor 1 (TNFR1) or Toll-like receptor 3 and 4 (TLR3/4) signalling. RIPK1 comprises an N-terminal kinase domain and is connected to a C-terminal death domain (DD) through an intermediate domain with a RIP homotypic interaction motif (RHIM) (He and Wang, 2018). Upon TNFR1 stimulation, RIPK1 is recruited to the cytoplasmic tail of TNFR1 via homotypic DD interaction, and RIPK1 acts as a scaffold to promote nuclear factor-κB (NF-κB) and mitogen-activated protein kinase (MAPK) activation, independent of its kinase activity (Newton, 2020). In addition, RIPK1 interacts with the TLR3/4 signalling adaptor, TIR domain-containing adapter inducing IFN-β (TRIF), through RHIM-RHIM interaction, where RIPK1 also serves as a scaffold to promote NF-κB and MAPK activation in TLR3/4-stimulated cells (Newton, 2020). Under both scenarios, kinases including transforming growth factor-β-activated kinase 1 (TAK1) and IκB kinase (IKK)α/β, as well as ubiquitin ligases like inhibitors of apoptosis proteins (IAPs) mediate inhibitory phosphorylation and ubiquitination of RIPK1 to maintain RIPK1's pro-survival function (Newton, 2020). Thus, under conditions where TNFR1 or TLR3/4 are engaged and inflammatory kinases or ubiquitin ligases are inactive, RIPK1 undergoes autophosphorylation and assembles a death-inducing complex which comprises the core components: RIPK1, Fas-associated protein with death domain (FADD) and caspase-8 (Dondelinger et al, 2013; Dondelinger et al, 2015). The scaffolding function of a second RHIM domain-containing protein, RIPK3, is additionally required for full caspase-8 activation in certain cell types such as murine embryonic fibroblasts and macrophages (Dondelinger et al, 2013; Lawlor et al, 2015; Vince et al, 2012).

RIPK1 kinase-dependent caspase-8 activation is a key anti-microbial mechanism to counteract pathogen blockade of innate immune signalling. For example, *Yersinia pestis* and *Yersinia pseudotuberculosis* (*Yptb*) use the T3SS effector, YopJ, to acetylate and inhibit TAK1, IKKα/β and MAPK kinases (MAPKKs) (Mukherjee et al, 2006). Coincident TNFR1 or TLR4 activation in the presence of YopJ unleashes RIPK1 kinase-driven caspase-8 activation (Peterson et al, 2016), where active caspase-8 cleaves executioner caspases-3 and -7 to drive extrinsic apoptosis (Philip et al, 2014; Weng et al, 2014). In macrophages, caspase-8 additionally cleaves gasdermin D (GSDMD) to release its cytotoxic N-terminal pore-forming fragment (p30), which permeabilises the

[1]Immunology Translational Research Programme, Department of Microbiology and Immunology, Yong Loo Lin School of Medicine, National University of Singapore, Singapore, Singapore. [2]Immunology Programme, Life Sciences Institute, National University of Singapore, Singapore, Singapore. [3]Centre for Innate Immunity and Infectious Diseases, Hudson Institute of Medical Research, Clayton, VIC, Australia. [4]Department of Molecular and Translational Science, Monash University, Clayton, VIC, Australia. [5]Department of Microbiology, Monash University, Clayton, VIC, Australia. [6]School of Medicine, University of St Andrews, St Andrews KY16 9TF Fife, UK. ✉E-mail: kaiwen.chen@nus.edu.sg

 

plasma membrane to elicit caspase-8-driven pyroptosis while caspase-3 cleaves GSDME to initiate pyroptosis in neutrophils (Chen et al, 2021; Demarco et al, 2020; Orning et al, 2018; Sarhan et al, 2018). Active caspase-8 also directly cleaves and activates the pro-inflammatory cytokine, interleukin (IL)-1β, and promotes NOD-like receptor protein 3 (NLRP3) inflammasome activation (Maelfait et al, 2008; Vince et al, 2012).

While caspase-8 drives potent pro-inflammatory and pro-death outcomes, it also plays a key role in restricting a third form of lytic cell death, known as necroptosis, by cleaving and inactivating two key RHIM domain-containing proteins, RIPK1 and RIPK3 (Kaiser et al, 2011; Lalaoui et al, 2020; Newton et al, 2019; Oberst et al, 2011; Tao et al, 2020). Thus, under conditions of caspase-8 blockade or deficiency, kinase-active RIPK1 recruits and activates RIPK3, which phosphorylates the pseudokinase, mixed lineage kinase domain-like pseudokinase (MLKL), to drive plasma membrane rupture and necroptosis (Murphy et al, 2013; Sun et al, 2012; Wang et al, 2012; Zhao et al, 2012). Based on these seminal findings, numerous studies over the past decade have used the combination of TNF or LPS, in conjunction with a TAK1/IKK/IAP inhibitor and caspase-8 inhibitor to interrogate necroptosis biology (Pasparakis and Vandenabeele, 2015). Although emerging studies document that sensing of Z-nucleic acid by a third RHIM domain-containing protein, Z-DNA-binding protein 1 (ZBP1), can directly engage RIPK3 to initiate necroptosis without engaging RIPK1 or its kinase activity (Jiao et al, 2020; Newton et al, 2016; Upton et al, 2012; Zhang et al, 2020), to the best of our knowledge, the physiological condition by which all three signals: (1) TNFR1 or TLR4 signalling, (2) TAK1/IKK inhibition, (3) caspase-8 blockade, are present during classical RIPK1/RIPK3-dependent necroptosis remains to be identified.

*Citrobacter rodentium* is a natural murine pathogen commonly used to model human ulcerative colitis and diarrheal disease caused by enteropathogenic and enterohaemorrhagic *Escherichia coli* (EPEC and EHEC respectively) infection (Collins et al, 2014). Upon oral ingestion, *C. rodentium* colonises the caecum, from where it translocates to the distal colon and adheres to colonic enterocytes by forming attaching and effacing (A/E) lesions (Collins et al, 2014). The formation of A/E lesions depends on the T3SS and distinguishes *C. rodentium*, EPEC and EHEC from other enteric pathogens. In addition, *C. rodentium*, EPEC and EHEC utilise T3SS effectors to subvert pro-inflammatory signalling and cell death responses. This includes effectors that inactivate the NF-κB pathway (NleB, NleC, NleE), caspase-8 (NleB, NleF) and RHIM proteins (EspL) (Blasche et al, 2013; Li et al, 2013; Nadler et al, 2010; Newton et al, 2010; Pearson et al, 2017; Pearson et al, 2013; Pearson et al, 2011; Pollock et al, 2017; Zhang et al, 2011). However, the interplay between these bacterial effectors and how the host counteracts their activities remain elusive.

In this study, we report *C. rodentium* as a physiologically relevant pathogen that fulfils all three requirements of classical necroptosis activation: (1) engagement of TLR4 and/or TNFR1 signalling, (2) TAK1/IKK blockade, and (3) caspase-8 inhibition. We demonstrate that bone marrow-derived macrophages (BMDMs) and murine colonic epithelial cells (CMT-93) retaliate against NleE blockade of innate immune signalling by triggering caspase-8 activation, and that caspase-8 activation is subverted by *C. rodentium* NleB. Although the inactivation of caspase-8 by NleB sensitises macrophages to necroptosis, *C.*

*rodentium* injects a third effector, EspL, to cleave and inactivate RIPK1 and RIPK3 to prevent necroptosis. Finally, we demonstrate that *C. rodentium* NleB and EspL collaborate to subvert caspase-8 activation and secondary NLRP3 activation. Thus, our study provides mechanistic insight into a bidirectional host-pathogen battle in regulating inflammation and cell death pathways.

# Results

## Macrophages and intestinal epithelial cells trigger caspase-8 activation in response to *C. rodentium* NleE blockade of innate signalling

We and others previously demonstrated that myeloid cells trigger RIPK1-dependent caspase-8 activation to counteract *Y. pseudotuberculosis* blockade of TAK1 or IKKα/β activity (Demarco et al, 2020; Orning et al, 2018; Peterson et al, 2017; Sarhan et al, 2018). Here, we examined whether this signalling axis protects against other pathogens that suppress innate immune signalling. First, we challenged bone marrow-derived macrophages (BMDMs) with wild-type *C. rodentium* or Δ*nleE* cultured under T3SS-inducing conditions. Unlike previous observations in epithelial cells (Nadler et al, 2010; Newton et al, 2010), the TAB2/3-targeting bacterial cysteine methyltransferase, NleE (Zhang et al, 2011), modestly suppressed NF-κB activation and induction of NF-κB target genes such as the anti-apoptotic protein MCL-1 (Speir et al, 2023) in BMDMs (Fig. EV1A,B). However, this amount of TAB2/3 inhibition is sufficient to elicit macrophage cell death since *C. rodentium*-induced lactate dehydrogenase (LDH) release (Fig. 1A), and apoptotic caspase-8 and -3 processing were significantly reduced in Δ*nleE*-infected BMDMs (Fig. 1B). Addition of the RIPK1 kinase inhibitor, Nec-1s, or genetic deletion of *Casp8* significantly reduced LDH release, apoptotic caspase processing and cleavage of downstream caspase substrates (Fig. 1A–D), suggesting that macrophages promote RIPK1 kinase-dependent caspase-8 activation to retaliate against NleE blockade of innate immune signalling during *C. rodentium* infection. Since *Casp8* deficiency unleashes MLKL activation and embryonic lethality (Alvarez-Diaz et al, 2016; Kaiser et al, 2011; Oberst et al, 2011), immortalised BMDMs (iBMDMs) from *Casp8*⁻/⁻*Mlkl*⁻/⁻ mice and *Mlkl*⁻/⁻ littermate controls were challenged with *C. rodentium* (Fig. 1C,D).

To delineate whether caspase-8 activation upon *C. rodentium* infection emanates from TNFR1 and/or TLR4-TRIF signalling, we prepared BMDMs from wild-type (WT), *Tnfr1*⁻/⁻, or *Trif*^Lps2 mice that contain a frameshift mutation and produce an unstable or non-functional TRIF mutant protein (Hoebe et al, 2003), and challenged these cells with *C. rodentium*. *Tnfr1* deficiency and *Trif*^Lps2 mutation protected macrophages from cell lysis at 3 h (Fig. EV1C) and 5 h (Fig. 1E) post-infection and these cells displayed reduced cleavage of caspase-8, caspase-3, GSDMD (Fig. 1F) as well as the fluorogenic caspase-3/7 substrate (Fig. EV1D) compared to WT macrophages. A recent study demonstrated that the TLR4 signalling adaptor, TRIF, promotes TNFR1-mediated cytokine production and cell death independent of TLR4 engagement (Muendlein et al, 2022). Although *Trif*^Lps2 BMDMs displayed impaired LPS-induced IRF3 phosphorylation compared to WT macrophages as anticipated (Fig. EV2A), we

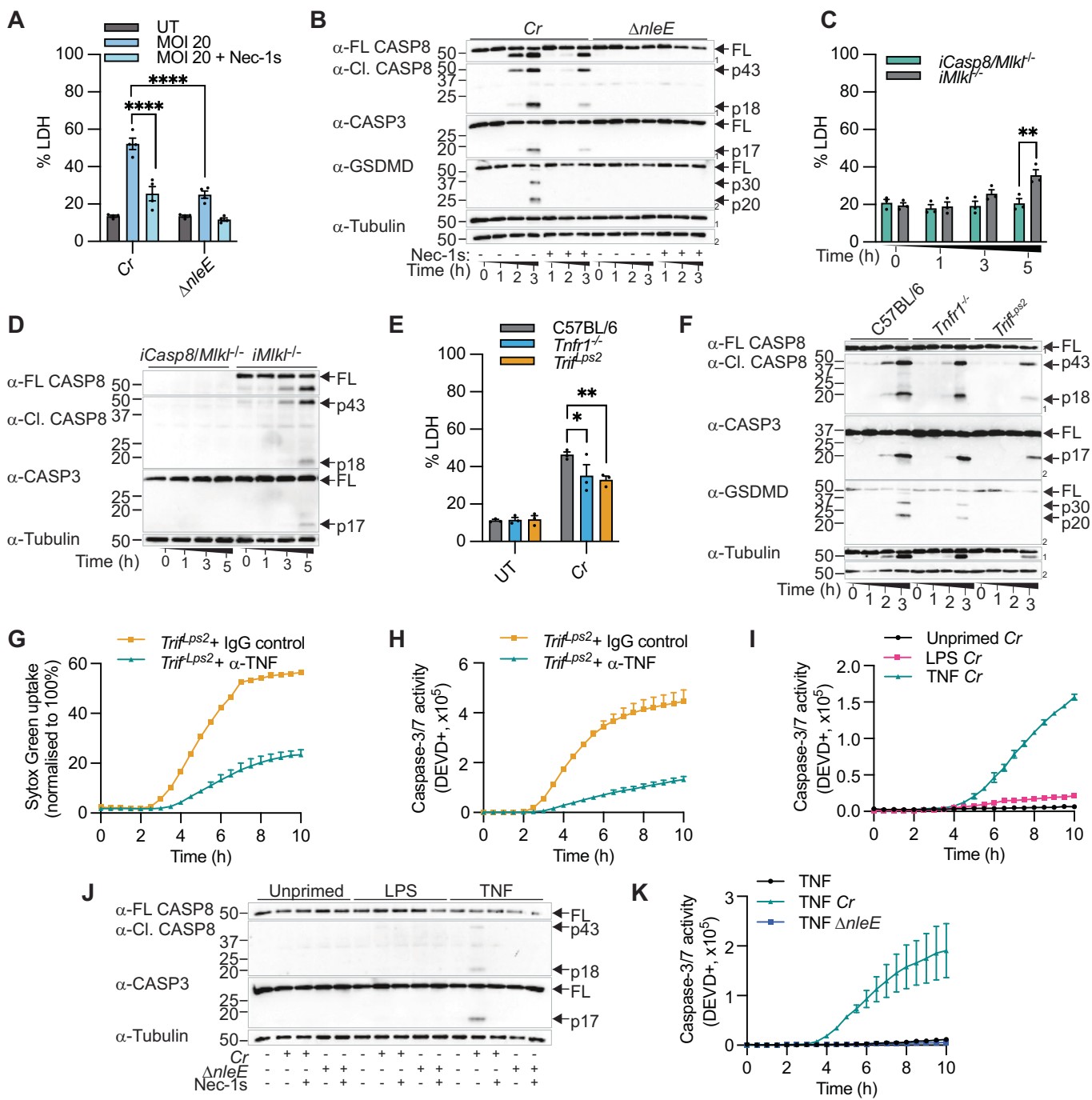

**Figure 1. Macrophages and colonic epithelial cells trigger caspase-8 activation in response to NleE upon *C. rodentium* infection.**

(A, B) Unprimed primary BMDMs, (C, D) immortalised BMDMs or (E–H) unprimed primary BMDMs were infected with log-phase *C. rodentium* (*Cr*) and/or Δ*nleE* for (A, E) 5 h or (B–D, F–H) the indicated time points. (G, H) BMDMs were treated with 100 µg/ml α-TNF neutralising antibody or IgG isotype control for 30 min before *Cr* infection. (I–K) CMT-93 cells were unprimed or primed with LPS (100 ng/ml) or TNF (100 ng/ml) for 3 h and infected with *Cr* and Δ*nleE* over time (I, K) or for (J) 5 h. (A, C, E) LDH release was quantified. (B, D, F, J) Mixed supernatant and lysates were examined by immunoblotting. (G) SYTOX Green uptake and (H, I, K) caspase-3/7 activity (DEVD-positive) were quantified using IncuCyte. Where indicated, cells were treated with Nec-1s (100 µM) for 30 min before infection. (A) Pooled data are mean ± SEM of four independent experiments (*P* ≤ 0.0001 for both conditions). (C) Pooled data are mean ± SEM of three independent experiments (*P* = 0.0010). (E) Pooled data are mean ± SEM of three independent experiments (*P* = 0.0257 for *Tnfr1*[−/−] and *P* = 0.0094 for *Trif*[Lps2]). Data are mean ± SD of technical (G, H) duplicates and (I, K) triplicates representative of three independent experiments. All *P* values were calculated with two-way ANOVA test. Data are considered significant when *P* ≤ 0.05, with **P* ≤ 0.05, ***P* ≤ 0.01 or *****P* ≤ 0.0001. Source data are available online for this figure.

observed that WT and *Trif*^Lps2^ macrophages displayed comparable NF-κB and MAPK activation (Fig. EV2B), CXCL2 secretion (Fig. EV2C), LDH release and apoptotic caspase processing (Fig. EV2D–F) following TNF stimulation under our experimental conditions. Moreover, TNF neutralisation in *Trif*^Lps2^ macrophages further reduced SYTOX Green uptake and cleavage of the fluorogenic caspase-3/7 substrate compared to *Trif*^Lps2^ macrophages treated with the IgG isotype control (Fig. 1G,H), suggesting that *C. rodentium* infection promotes macrophage death through two parallel pathways that are driven by TNFR1 and TLR4-TRIF respectively.

Next, to examine whether RIPK1 kinase-mediated cell death also protects colonic epithelial cells against *C. rodentium* blockade of innate immune signalling, we challenged the murine colorectal cancer cell line, CMT-93, with *C. rodentium* and *ΔnleE* and verified that NleE likewise suppresses NF-κB activation and induction of the NF-κB target gene, c-FLIP, in these cells (Appendix Fig. S1A,B). Interestingly, unlike in macrophages (Fig. 1A,B), *C. rodentium* infection did not trigger apoptotic caspase activation in unprimed CMT-94 cells (Fig. 1I,J). Instead, TNF priming was required to promote caspase-3/7 activity, with apoptotic caspase cleavage detectable from 5 h post-infection (Fig. 1I,J). Consistent with macrophage infection (Fig. 1A,B), apoptotic caspase activation in TNF-primed CMT-93 cells was diminished following Nec-1s treatment or *ΔnleE* infection (Fig. 1J,K). LPS priming triggered weak caspase-3/7 activity following *C. rodentium* infection in CMT-93 cells (Fig. 1I), but apoptotic caspase processing observed by immunoblot was below the detection limit (Fig. 1J). Hence, we used TNF-primed CMT-93 cells as a model to study *C. rodentium* infection in colonic epithelial cells for the remainder of the study. In summary, we uncovered apoptotic caspase activation in macrophages and colonic epithelial cells as a host response to counteract NleE subversion of innate immune signalling.

## Loss of EspL alone does not promote apoptosis in macrophages or epithelial cells during *C. rodentium* infection

Our results thus far suggest RIPK1-dependent caspase-8 activation as a conserved host protective mechanism upon pathogen blockade of innate immune signalling. However, we observed that *C. rodentium* triggered weaker macrophage lysis (Fig. 2A,B) and slower kinetics of apoptotic caspase activation than *Y. pseudotuberculosis*-infected macrophages (Fig. 2C,D; Appendix Fig. S2A). These led us to hypothesise that *C. rodentium* translocates T3SS effectors to subvert apoptotic signalling. We first examined whether the bacterial cysteine protease effector, EspL, which was previously reported to cleave and inactivate RHIM domain-containing proteins in human intestinal epithelial cells (Pearson et al, 2017), also inactivates RIPK1 and RIPK3 in primary murine macrophages. To do so, we infected BMDMs with *C. rodentium* or *ΔespL* in the presence of the pan-caspase inhibitor, Q-VD-Oph (QVD), to suppress RIPK1 and RIPK3 cleavage by apoptotic caspases (Kaiser et al, 2011; Lalaoui et al, 2020; Newton et al, 2019; Oberst et al, 2011; Tao et al, 2020). Although EspL cleaved RIPK1 and RIPK3 to generate the expected p61 and p50 fragments (Fig. 2E; Appendix Fig. S2B), LDH release, apoptotic caspase processing and cleavage of the fluorogenic caspase substrates were comparable between *C. rodentium* and *ΔespL* infection in macrophages and CMT-93 cells

(Fig. 2F–I). Collectively, these data suggest that loss of EspL alone does not significantly promote apoptosis in macrophages or epithelial cells.

## EspL subverts macrophage necroptosis

Although loss of EspL alone did not enhance macrophage or colonic epithelial cell apoptosis (Fig. 2F–I), *ΔespL* was attenuated for colonic intestinal persistence between 12 to 18 DPI (Fig. EV3A), as previously shown in C57BL/6 mice (Pearson et al, 2017). Since RHIM domain-containing proteins are also key drivers of necroptosis, we reasoned that EspL promotes microbial persistence by subverting necroptosis. While *ΔespL* triggered a subtle increase in MLKL phosphorylation compared to wild-type *C. rodentium* infection in unprimed macrophages (Fig. 3A), LDH release following *ΔespL* infection remains comparable (Fig. 3B) between unprimed WT and the necroptosis-deficient *Mlkl*^SA2^ macrophages (EV3B,C), in which the two key RIPK3 phosphorylation sites (S345/347) on MLKL were replaced with alanine. Next, we primed macrophages with LPS to induce the expression of anti-apoptotic proteins to suppress caspase-8 activity and favour necroptosis, and first confirmed that EspL cleaved RIPK1 and RIPK3 in LPS-primed macrophages (Fig. EV3D). Indeed, LPS priming suppressed *C. rodentium*- and *ΔespL*-induced caspase-8 and caspase-3 processing, but significantly enhanced *ΔespL*-induced MLKL phosphorylation compared to unprimed WT macrophages (Fig. 3A). In contrast, MLKL phosphorylation was not detected following wild-type *C. rodentium* infection in WT macrophages or following *ΔespL* infection in *Mlkl*^SA2^ macrophages (Fig. 3A), as anticipated. This mode of *ΔespL*-mediated cell death was significantly reduced in LPS-primed *Mlkl*^SA2^ and *Ripk3*^−/−^ macrophages compared to WT macrophages (Fig. 3B,C), and was sensitive to Nec-1s and the RIPK3 kinase inhibitor, GSK'872 (Fig. 3D), altogether suggesting that *ΔespL* triggers necroptosis in LPS-primed macrophages. Previous studies reported that active MLKL promotes NLRP3 assembly in a cell-intrinsic manner (Conos et al, 2017; Gutierrez et al, 2017). Consistent with this, *ΔespL* but not *C. rodentium* triggered RIPK3- and NLRP3-dependent caspase-1 processing (Fig. 3E,F) and RIPK3-dependent IL-1β secretion (Fig. 3G). Taken together, our findings suggest that *ΔespL* infection predominantly triggers apoptosis in majority of unprimed macrophages and necroptosis in a small population of macrophages, while LPS priming suppresses apoptosis and sensitises macrophages to necroptosis.

Next, to elucidate whether EspL subverts necroptosis in murine colonic epithelial cells, we first profiled the expression of necroptotic factors in naïve CMT-93 cells and after 24 h priming with LPS, interferon (IFN)-γ, IL-1α or IL-1β. We excluded TNF priming because prolonged TNF treatment (>10 h) in CMT-93 cells triggered spontaneous caspase-3/7 activity in the absence of infection (Fig. EV3E). RIPK1 and MLKL were readily expressed in unprimed CMT-93 cells, while IFN-γ priming further induced MLKL expression (Fig. 3H). However, RIPK3 remained weakly expressed in unprimed cells and after 24 h priming with LPS, IFN-γ, IL-1α or IL-1β (Fig. 3H). Thus, we generated a RIPK3-expressing stable CMT-93 cell line (Fig. EV3F) that is susceptible to TNF and emricasan-induced necroptosis (Fig. EV3G). However, both wild-type *C. rodentium* and *ΔespL* infection triggered comparable amounts of cell death in RIPK3-expressing CMT-93 cells and this

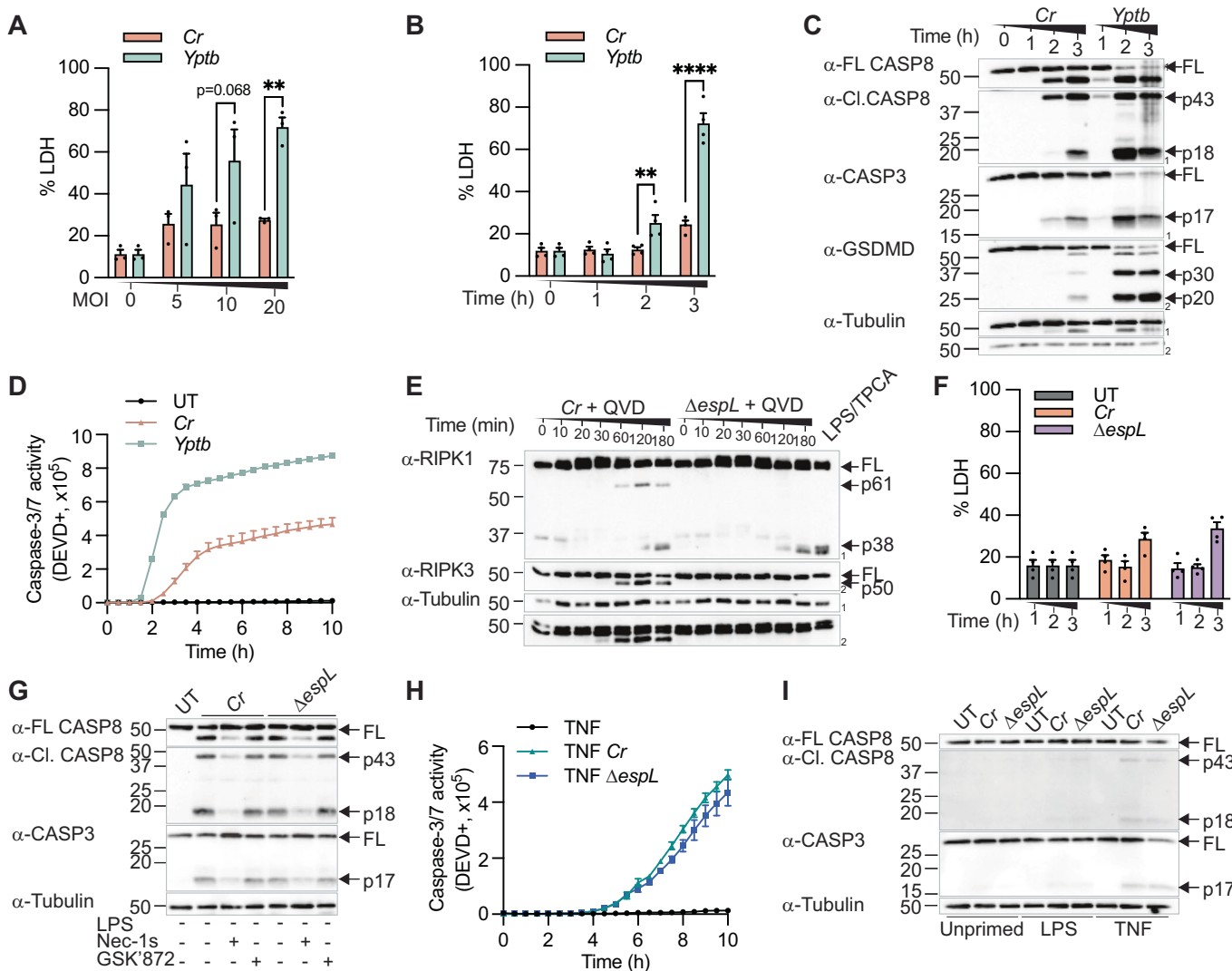

**Figure 2. Loss of EspL alone does not promote apoptosis in macrophages or colonic epithelial cells infected with *C. rodentium*.**

(A–D) Unprimed BMDMs were infected with log-phase *C. rodentium* (*Cr*) or *Y. pseudotuberculosis* (*Yptb*) for (A) 3 h or (B–D) the indicated time points. (E–G) Unprimed BMDMs and (H, I) CMT-93 cells primed with LPS (100 ng/ml) or TNF (100 ng/ml) were infected with *Cr* or *ΔespL* for (E, F, H) the indicated time points, (G) 3 h or (I) 5 h. (A, B, F) LDH release was quantified. (C, E, G, I) Mixed supernatant and lysates were examined by immunoblotting. (E) BMDMs co-stimulated with LPS (100 ng/ml) and TPCA (5 μM) as positive control showing caspase-8 active cleavage of RIPK1. (D, H) Caspase-3/7 activity (DEVD-positive) was quantified using IncuCyte. Where indicated, cells were treated with QVD (20 μM), Nec-1s (100 μM) or GSK'872 (5 μM) for 30 min before infection. (A) Pooled data are mean ± SEM of three independent experiments (*P* = 0.0056 for MOI 20). (B) Pooled data are mean ± SEM of four independent experiments (*P* = 0.0068 for 2 h and *P* ≤ 0.0001 for 3 h). (F) Pooled data are mean ± SEM of four independent experiments. Data are mean ± SD of (D, H) technical duplicates representative of two independent experiments. All *P* values were calculated with two-way ANOVA test. Data are considered significant when *P* ≤ 0.05, with \*\**P* ≤ 0.01 or \*\*\*\**P* ≤ 0.0001. Source data are available online for this figure.

death was insensitive to GSK'872 treatment (Fig. 3I). These findings suggest that failure to cleave RHIM-containing proteins by EspL does not sensitise RIPK3-expressing stable CMT-93 cells to necroptosis during *ΔespL* infection and prompted us to investigate the underlying mechanism for necroptosis activation in LPS-primed macrophages.

## TRIF but not TNFR1 promotes macrophage necroptosis upon *ΔespL* infection

LPS stimulation induces autocrine TNF and type I IFN signalling, which may license macrophages to *ΔespL*-induced necroptosis by promoting

TNFR1-dependent cell death, or transcriptional induction of necroptotic machineries such as *Mlkl* (Thapa et al, 2013), respectively. To examine these possibilities, we prepared LPS-primed WT, *Tnfr1*[−/−], and *Trif*[Lps2] macrophages and challenged these cells with *C. rodentium* and *ΔespL*. While *ΔespL* infection triggered comparable MLKL phosphorylation between WT, *Tnfr1*[−/−] and *Trif*[Lps2] macrophages (Fig. 4A), caspase-1 processing, GSDMD cleavage (Fig. 4A) and macrophage lysis (Fig. 4B) were reduced in *Trif*[Lps2] macrophages compared to WT and *Tnfr1*[−/−] macrophages. IL-1β secretion from *ΔespL*-infected *Trif*[Lps2] macrophages was close to negligible (Fig. 4C), likely due to the compounded effect of impaired pro-IL-1β expression (Yow et al, 2024) and defective caspase-1-dependent IL-1β maturation (Fig. 4A).

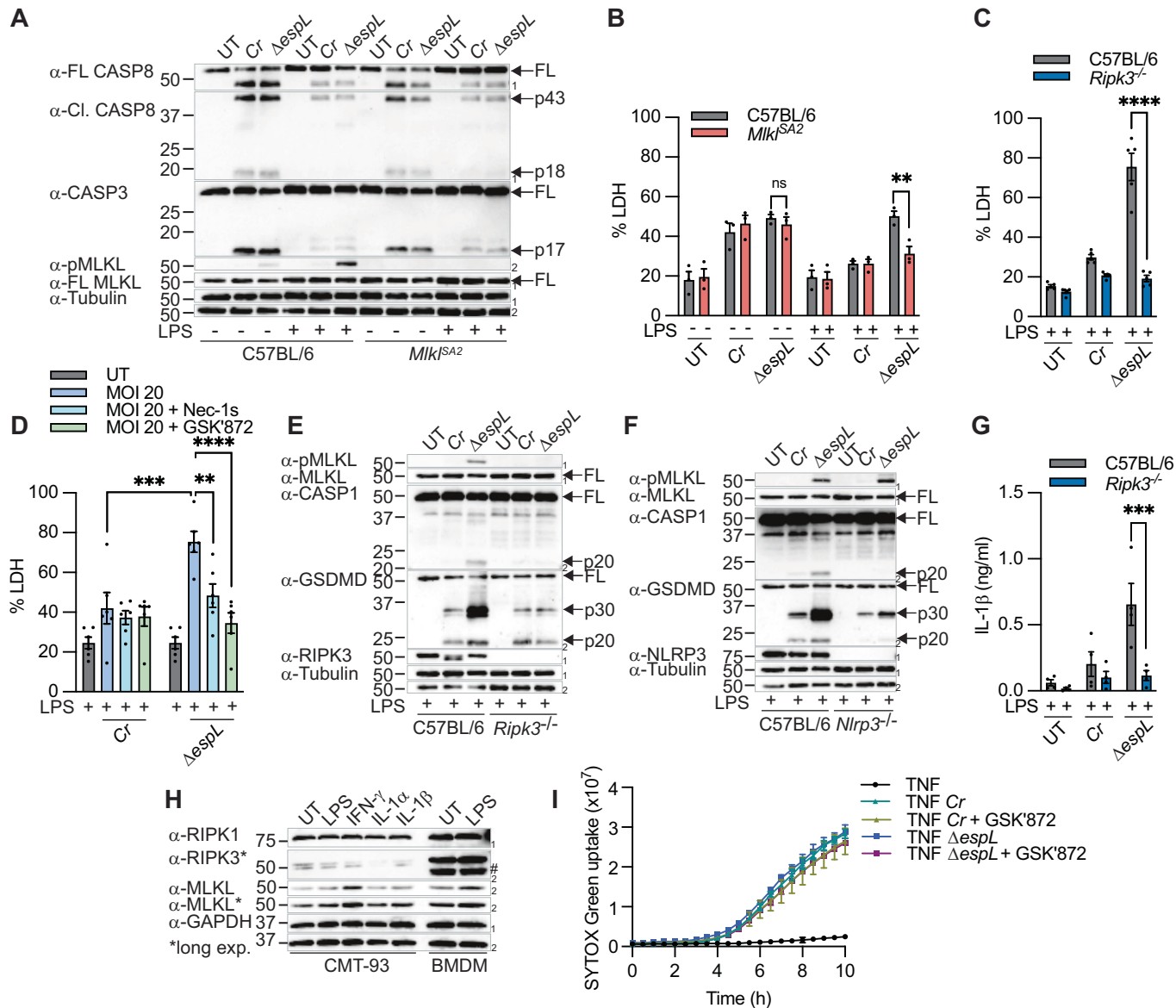

**Figure 3. EspL subverts macrophage necroptosis.**

(**A–G**) BMDMs were unprimed or primed with LPS (100 ng/ml) for 3 h and challenged with log-phase *C. rodentium* (*Cr*) or Δ*espL* for 5 h. (**H**) CMT-93 cells were either unprimed or primed with LPS (100 ng/ml), IFN-γ (100 ng/ml), IL-1α (100 ng/ml) or IL-1β (100 ng/ml) for 24 h. Unprimed and LPS-primed (100 ng/ml) BMDMs were included as positive controls. (**I**) RIPK3-expressing CMT-93 cells were treated with TNF (100 ng/ml) upon *Cr* or Δ*espL* infection and SYTOX Green was quantified using IncuCyte. (**B–D**) LDH and (**G**) IL-1β release were measured. (**A, E, F**) Mixed supernatant and lysates or (**H**) cell lysates were examined by immunoblotting. Where indicated, cells were treated with Nec-1s (100 μM) or GSK'872 (5 μM) for 30 min before infection. (**B**) Pooled data are mean ± SEM of three independent experiments (*P* = 0.0038). (**C**) Pooled data are mean ± SEM of five independent experiments (*P* ≤ 0.0001). (**D**) Pooled data are mean ± SEM of six independent experiments (*P* = 0.0002 for *Cr* vs Δ*espL*, *P* = 0.0034 for Δ*espL* treated with Nec-1s and *P* ≤ 0.0001 for Δ*espL* treated with GSK'872). (**G**) Pooled data are mean ± SEM of four independent experiments (*P* = 0.0004). Data are mean ± SD of (**I**) technical duplicates representative of three independent experiments. All *P* values were calculated with two-way ANOVA test. Data are considered significant when *P* ≤ 0.05, with \*\**P* ≤ 0.01, \*\*\**P* ≤ 0.001 or \*\*\*\**P* ≤ 0.0001. #non-specific band. Source data are available online for this figure.

Next, we challenged LPS-primed WT and *Ifnar1*⁻/⁻ macrophages with Δ*espL* to determine whether TRIF promotes macrophage necroptosis by inducing autocrine type I IFN signalling. Unexpectedly, Δ*espL* infection triggered comparable GSK'872-sensitive macrophage cytotoxicity and IL-1β release between WT and *Ifnar1*⁻/⁻ macrophages (Fig. 4D,E), suggesting that autocrine type I IFN signalling is dispensable for Δ*espL*-induced necroptosis.

In support of this, priming macrophages with the synthetic TLR1/2 agonist, Pam3CSK4, that exclusively engages MyD88 and not TRIF signalling to induce expression of anti-apoptotic proteins, likewise sensitised macrophages to TRIF-dependent necroptosis following Δ*espL* infection (Figs. 4F and EV4A). Since RIPK3 and MLKL expression is comparable between WT and *Trif*^Lps2 macrophages before or after LPS or TNF stimulation (Fig. EV4B), our data

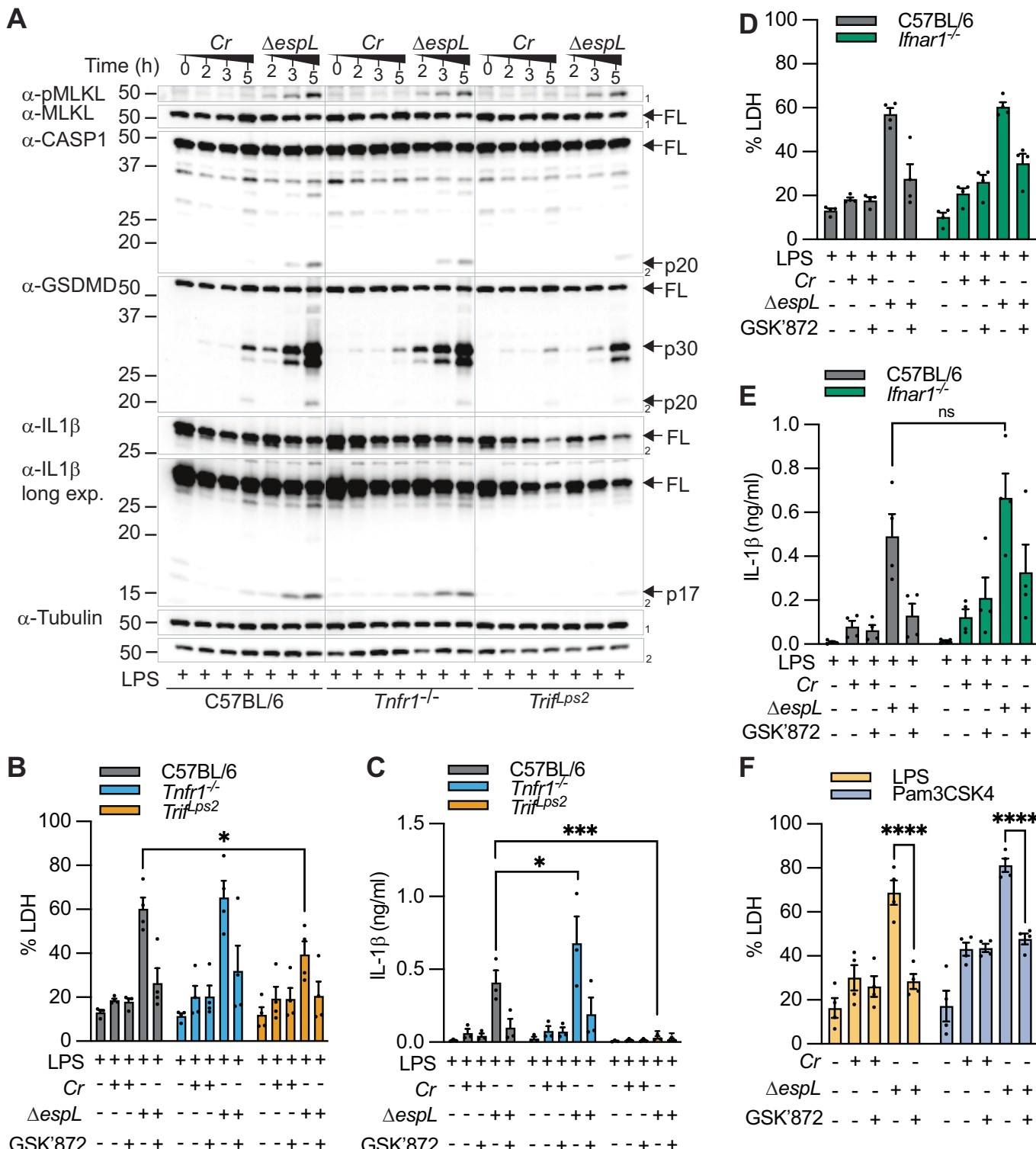

**Figure 4. ΔespL promotes macrophage necroptosis through TRIF and not TNFR1 signalling.**

BMDMs were primed with (**A–E**) LPS (100 ng/ml) or (**F**) Pam3CSK4 (1 μg/ml) for 3 h before infected with log-phase *C. rodentium* (*Cr*) and Δ*espL* for (**A**) the indicated time points or (**B–F**) 5 h. (**A**) Mixed supernatant and lysates were examined by immunoblot. (**B, D, F**) LDH and (**C, E**) IL-1β release were measured. Where indicated, cells were treated with GSK'872 (5 μM) for 30 min before infection. (**B**) Pooled data are mean ± SEM of four independent experiments (*P* = 0.0213). (**C**) Pooled data are mean ± SEM of three independent experiments (*P* = 0.0106 for Δ*espL* infection in *Tnfr1⁻/⁻* and *P* = 0.0005 for *Trif^Lps2*). (**D, E**) Pooled data are mean ± SEM of four independent experiments. (**F**) Pooled data are mean ± SEM of four independent experiments (*P* ≤ 0.0001 for both conditions) independent experiments. All *P* values were calculated with two-way ANOVA test. Data are considered significant when *P* ≤ 0.05, with *\*P* ≤ 0.05, *\*\*\*P* ≤ 0.001 or *\*\*\*\*P* ≤ 0.0001. Source data are available online for this figure.

suggest that TRIF promotes Δ*espL*-induced necroptosis through its scaffolding function (He et al, 2011; Kaiser et al, 2013) but not via autocrine type I IFN signalling and transcriptional induction of RIPK3 or MLKL.

## *C. rodentium* EspL counteracts NleB blockade of caspase-8 to suppress necroptosis in macrophages

Our data demonstrate that *C. rodentium* requires EspL to inactivate RHIM-containing proteins to subvert macrophage necroptosis (Fig. 3A–C). Since necroptosis occurs upon caspase-8 deficiency, this suggests that additional *C. rodentium* T3SS effector(s) subvert caspase-8 activation in macrophages. Thus, we examined whether NleB and NleF, which were previously reported to suppress caspase-8 activation in epithelial cells (Blasche et al, 2013; Li et al, 2013; Pearson et al, 2017; Pollock et al, 2017), also suppress caspase-8 activation in macrophages. Δ*nleB* but not Δ*nleF* elicited significantly higher LDH release and apoptotic caspase activation compared to wild-type *C. rodentium* in unprimed macrophages (Fig. 5A,B), suggesting that NleB has a dominant role in suppressing macrophage caspase activation, while loss of NleF alone has limited impact on macrophage death (Fig. 5A,B).

To investigate whether NleB inhibition of caspase-8 signalling sensitises macrophages to necroptosis in the absence of EspL, we deleted *nleB* from the Δ*espL* *C. rodentium* strain to generate a Δ*nleB/espL* double mutant. Indeed, Δ*nleB/espL* triggered reduced MLKL phosphorylation compared to Δ*espL* following infection in LPS-primed macrophages (Fig. 5C), and LDH release by Δ*nleB/espL* was no longer sensitive to GSK'872 inhibition (Fig. 5D). Additionally, we observed that Δ*nleB/espL* triggered enhanced macrophage cytotoxicity, caspase-8 and caspase-1 processing, GSDMD cleavage and IL-1β maturation compared to Δ*nleB* infection in LPS-primed macrophages (Fig. 5D,E), while Δ*nleB/espL* complemented with *nleB* or *espL* caused less cell lysis (Fig. EV5A). The increase in macrophage cytotoxicity and inflammasome activation in Δ*nleB/espL*-infected macrophages require caspase-8 (Figs. 5F and EV5B) and is sensitive to Nec-1s and the NLRP3 inflammasome inhibitor, MCC950 (Fig. 5E), suggesting that EspL suppresses caspase-8-dependent NLRP3 activation in the absence of NleB.

Next, we generated a Δ*nleF/espL* double mutant to experimentally validate whether NleF sensitises Δ*espL*−infected macrophages to necroptosis. Consistent with our earlier observation (Fig. 5A,B), loss of NleF alone did not suppress Δ*espL*-induced necroptosis, since macrophage lysis following Δ*nleF/espL* infection remains sensitive to GSK'872 inhibition (Fig. EV5C), and MLKL phosphorylation upon Δ*espL* or Δ*nleF/espL* infection was identical (Fig. EV5D). Previous studies demonstrated that stationary phase *C. rodentium*, which does not favour T3SS expression, activates caspase-11, the apical caspase in the non-canonical inflammasome pathway (Fig. EV5E) (Gurung et al, 2012; Rathinam et al, 2012). We confirmed that *C. rodentium* grown to log phase that favours the expression of T3SS and its effectors and trigger caspase-11-independent macrophage cell death, apoptotic caspase and inflammasome activation (Fig. EV5F–H). Lastly, we investigated whether NleB and EspL together suppress apoptosis in CMT-93 cells. We observed that Δ*nleB* triggered enhanced caspase-8 and caspase-3 cleavage in CMT-93 cells compared to wild-type *C. rodentium* (Fig. 5G), indicating that NleB has an important role in

subverting caspase-8 activation in both macrophages and epithelial cells. However, unlike in macrophages, the Δ*nleB/espL* double mutant triggered comparable apoptotic caspase activation compared to Δ*nleB* infection (Fig. 5G). In summary, our findings highlight a complex interplay between NleE, NleB and EspL that enables *C. rodentium* to subvert innate immune signalling, apoptosis, necroptosis and NLRP3 activation in epithelial cells and macrophages.

## Discussion

We and others previously demonstrated that macrophages and neutrophils activate RIPK1 kinase-dependent caspase-8 activation as a key anti-bacterial strategy to overcome YopJ blockade of pro-inflammatory gene expression during *Y. pseudotuberculosis* infection (Chen et al, 2021; Demarco et al, 2020; Peterson et al, 2017; Philip et al, 2014; Weng et al, 2014). In this study, we report that macrophages and colonic epithelial cells respond to NleE blockade of innate immune signalling by inducing RIPK1 kinase-dependent caspase-8 activation during *C. rodentium* infection (Appendix Fig. S3A,C), suggesting that this signalling axis is a conserved host response in various cell types and is likely to protect against a variety of pathogens that suppress TAB2/3, TAK1 or IKK signalling.

Blockade of TAK1 or IKK signalling downstream of TNFR1 or TLR4 engagement promotes the assembly of two distinct caspase-8 signalling complexes, known as Complex IIb (Wang et al, 2008) and the TRIFosome (Muendlein et al, 2021) respectively. A recent study reported that the TLR3/4 signalling adaptor, TRIF, promotes TNFR1-dependent inflammation and cell death, independent of TLR3/4 engagement (Muendlein et al, 2022), highlighting the complexity and crosstalk between these cell death pathways. Although we observed that WT and *Trif*^*Lps2* macrophages displayed comparable TNF-dependent inflammation and cell death, our study does not rule out the possibility that TRIF is involved in TNFR1-driven processes, since the *Trif*^*Lps2* macrophages used in our study may still express a truncated TRIF protein that can promote TNF signalling. Future studies using *Trif* knockouts will be useful to further characterise the molecular functions of TRIF in TNFR1 signalling in cells of human and mouse origins and in different cell types.

Apoptotic caspase activation promotes a myriad of anti-microbial responses including NLRP3 inflammasome activation and induction of Th17 immunity (Chen et al, 2019; Lawlor et al, 2017; Torchinsky et al, 2009; Vince et al, 2012) Thus, it is reasonable to assume that pathogens have co-evolved multiple subversion strategies to promote microbial colonisation and persistence. One such candidate is EspL, a bacterial cysteine protease that cleaves RHIM domain-containing proteins including RIPK1 and RIPK3 (Pearson et al, 2017). Given that RIPK1 dimerisation promotes caspase-8 activation (Meng et al, 2018), and RIPK3 licenses apoptotic caspase activation downstream of TNFR1 and TLR4 (Chen et al, 2019; Dondelinger et al, 2013; Vince et al, 2012), we hypothesised that EspL subverts apoptosis by cleaving RIPK1 and RIPK3. However, we observed that loss of EspL alone had minimal impact on the kinetics or magnitude of apoptosis in macrophages and colonic epithelial cells. Instead, EspL was required to suppress necroptosis and secondary NLRP3

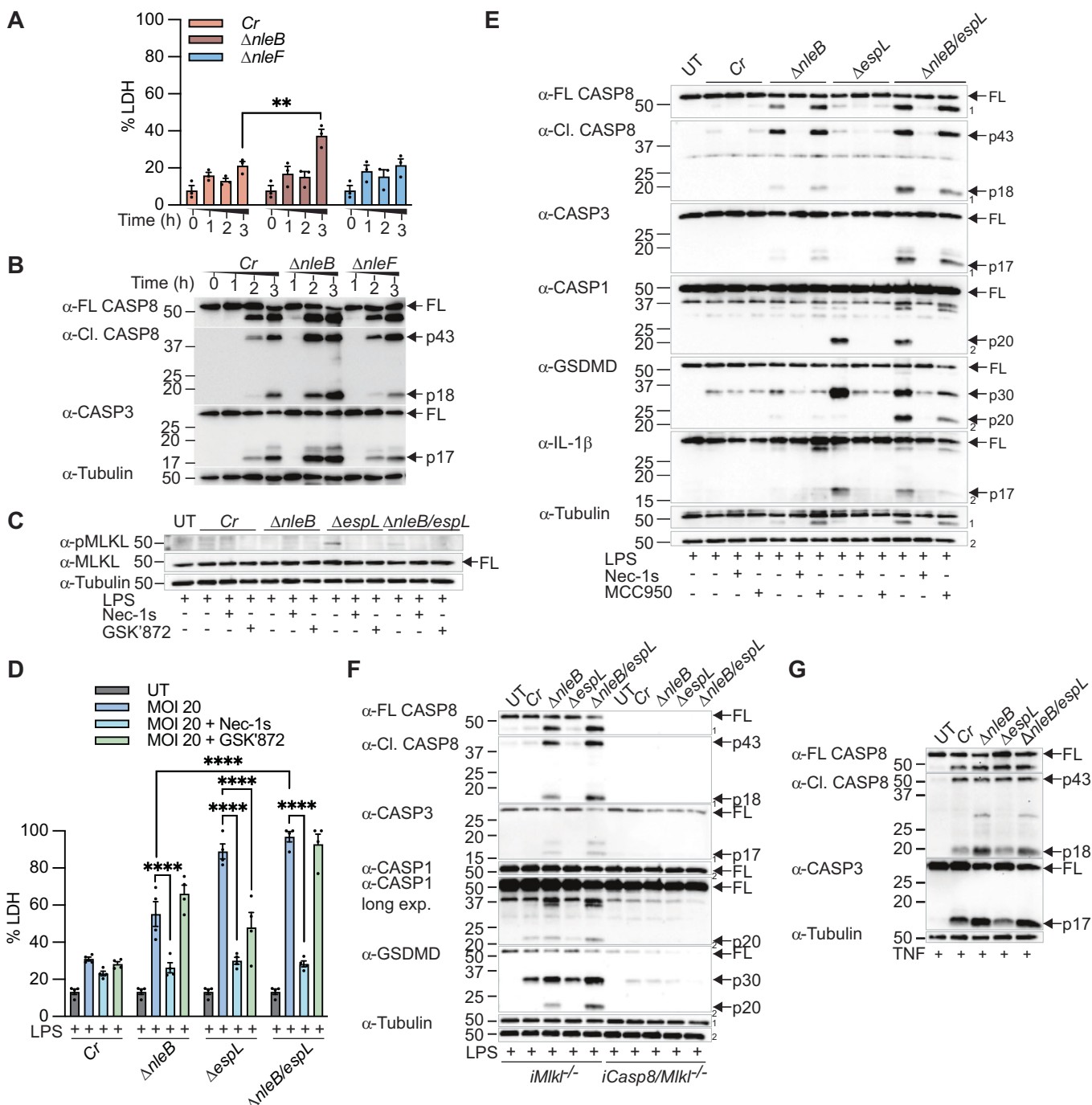

**Figure 5. EspL suppresses necroptosis in response to NleB blockade of caspase-8.**

(A, B) Unprimed BMDMs were infected with log-phase *C. rodentium* (*Cr*), Δ*nleB* or Δ*nleF* for the indicated time points. (C–E) BMDMs were primed with LPS (100 ng/ml) before infected with log-phase *Cr*, Δ*nleB*, Δ*espL* or Δ*nleB/espL* for (C, E) 3 h or (D) 5 h. (F) Immortalised BMDMs were primed with LPS (500 ng/ml) before infected with log-phase *Cr*, Δ*nleB*, Δ*espL* or Δ*nleB/espL* for 5 h. (G) CMT-93 cells were primed with TNF (100 ng/ml) for 3 h before infected with log-phase *Cr*, Δ*nleB*, Δ*espL* or Δ*nleB/espL* for 5 h. (A, D) LDH release was measured. (B, C, E–G) Mixed supernatant and lysates were examined by immunoblotting. Where indicated, cells were treated with Nec-1s (100 µM), GSK'872 (5 µM) or MCC950 (10 µM) for 30 min before infection. (A) Pooled data are mean ± SEM of three independent experiments (*P* = 0.0016). (D) Pooled data are mean ± SEM of four independent experiments (*P* ≤ 0.0001 for all five statistically significant conditions). All *P* values were calculated with two-way ANOVA test. Data are considered significant when *P* ≤ 0.05, with **\*\*P* ≤ 0.01, or *\*\*\*\*P* ≤ 0.0001. Source data are available online for this figure.

inflammasome activation in TLR1/2 or TLR4-primed macrophages (Appendix Fig. S3B). Given that caspase-8 blockade is often the pre-requisite for initiating necroptosis, we subsequently found that NleB is sufficient to suppress apoptosis and this sensitises Pam3CSK4 or LPS-primed macrophages to necroptosis unless EspL is present. In addition, deletion of NleB unleashes the ability of EspL to subvert apoptosis and secondary NLRP3 inflammasome activation (Appendix Fig. S3B). Thus, our study provides, at least in part, a mechanistic explanation of why machine learning algorithms predicted the NleB and EspL combination as interacting partners (Ruano-Gallego et al, 2021). Notably, NleE, NleB and EspL, through evolutionary pressure, are encoded on the same pathogenicity island (Petty et al, 2010) and are translocated hierarchically, starting from NleE, followed by NleB to EspL (Mills et al, 2013), altogether demonstrating how the intricate interplay between NleE, NleB and EspL affects host cell susceptibility to inflammation and cell death.

Unlike macrophages, we observed that Δ*espL* did not trigger necroptosis in CMT-93 colonic epithelial cells, even when these cells stably expressed RIPK3 and were sensitive to TNF/emricasan-induced necroptosis. This suggests that CMT-93 cells lack expression of key anti-apoptotic factors that suppress caspase-8 activation and sensitise CMT-93 cells to necroptosis during Δ*espL* infection, akin to our observations in unprimed macrophages. Nonetheless, our study does not exclude the possibility that Δ*espL* triggers colonic epithelial necroptosis in vivo, since several studies have concluded that intestinal epithelial cells are proficient in RIPK3- or MLKL-driven cell death and intestinal inflammation when *Casp8* is ablated (Gunther et al, 2011; Schwarzer et al, 2020; Welz et al, 2011; Wittkopf et al, 2013). Given that our kinetic analysis and previous work (Pearson et al, 2017) have shown that Δ*espL* is only attenuated for intestinal persistence during the clearance phase of infection that coincides with immune cells recruitment to the colon (Crepin et al, 2016), this points towards the possibility that the immune population is a possible target of EspL in vivo. As an A/E pathogen, *C. rodentium* would benefit from suppressing cell death. However, the additional suppression of NLRP3 inflammasome activation (Conos et al, 2017; Gutierrez et al, 2017; Kaczmarek et al, 2013) in necroptotic cells is likely a key microbial evasion mechanism. Future work assessing which immune subsets are targets of EspL during *C. rodentium* infection in vivo is crucial to unveil how necroptosis confers anti-bacterial defence.

Our observations and previous studies elucidating how EspL in A/E pathogens suppress necroptosis (Pearson et al, 2017) appear as a common mechanism by which enteric pathogens counteract pathogen blockade of caspase-8 activity. For example, *Shigella flexneri* similarly uses the EspL homologue, OspD3, to cleave RIPK1/3 and suppress necroptosis upon OspC1 blockade of capase-8 in human epithelial cells (Ashida et al, 2020). However, it remains unclear how caspase-8 is activated during *S. flexneri* infection and whether this also occurs in immune cells. If so, whether OspD3 and OspC1 collaborate to prevent apoptosis and downstream NLRP3 activation in macrophages, similar to our findings for EspL and NleB, remains to be determined. Another notable example occurs during cytomegalovirus (CMV) infection, where CMV uses viral M45-encoded inhibitor of RIP activation (vIRA) to disrupt RIPK1-RIPK3 interaction to counteract necroptosis upon caspase-8 blockade by viral inhibitor of caspase-8 activation (vICA) (Upton et al, 2010). Taken together, this study has mechanistically revealed

how epithelial cells or macrophages counteract the *C. rodentium* T3SS effectors that block apoptosis and necroptosis to elicit host immune defence. Future work leveraging this axis may be beneficial for uncovering new therapeutics for infectious and inflammatory diseases.

# Methods

## Reagents and tools table

| Reagent/resource | Reference or source | Identifier or catalog number |
|---|---|---|
| **Experimental models** | | |
| C57BL/6J (*M. musculus*) | Jackson Lab | #000664; RRID: IMSR_JAX:000664 |
| C57BL/6-*Tnfrsf1a*tm1Imx/J (*M. musculus*) | Jackson Lab | #003242; RRID: IMSR_JAX:003242 |
| C57BL/6J-*Ticam1*Lps2/J (*M. musculus*) | Jackson Lab | #005037; RRID: IMSR_JAX:005037 |
| C57BL/6N-*Gsdmd*em4Fcw/J (*M. musculus*) | Jackson Lab | #032410; RRID: IMSR_JAX:032410 |
| B6.129S4(D2)-*Casp4*tm1Yuan/J (*M. musculus*) | Jackson Lab | #024698; RRID: IMSR_JAX:024698 |
| B6.129S6-*Nlrp3*tm1Bhk/J (*M. musculus*) | Jackson Lab | #021302; RRID: IMSR_JAX:021302 |
| C57BL/6N-*Mlkl*em1Najaf/J (*M. musculus*) | Jackson Lab | #039024; RRID:IMSR_JAX:039024 |
| B6.129S2-*Ifnar1*tm1Agt/Mmjax (*M. musculus*) | Jackson Lab | #032045-JAX; RRID: MMRRC_032045-JAX |
| *Ripk3*−/− (*M. musculus*) | Newton et al, 2004 | |
| *Casp8*−/−*Mlkl*−/− (*M. musculus*) | Alvarez-Diaz et al, 2016 | |
| *Mlkl*−/− (*M. musculus*) | Murphy et al, 2013 | |
| *Casp8*−/−*Mlkl*−/− iBMDM | This study | |
| *Mlkl*−/− iBMDM | This study | |
| CMT-93 cells (*M. musculus*) | ATCC | RRID: CCL-223 |
| Stable RIPK3-expressing CMT-93 cells | This study | |
| *Citrobacter rodentium* ICC169, Nal^R | Mundy et al, 2003 | |
| *Citrobacter rodentium* Δ*espL*, Kan^R | Pearson et al, 2017 | |
| *Citrobacter rodentium* Δ*nleE*, Kan^R | Kelly et al, 2006 | |
| *Citrobacter rodentium* Δ*nleB*, Kan^R | Pearson et al, 2013 | |
| *Citrobacter rodentium* Δ*nleF*, Kan^R | Pallett et al, 2017 | |
| *Citrobacter rodentium* Δ*nleB*/*espL*, Kan^R/Cm^R | This study | |
| *Citrobacter rodentium* Δ*nleF*/*espL*, Kan^R/Cm^R | This study | |
| *Citrobacter rodentium* Δ*nleB*/*espL* (*nleB*), Kan^R/Cm^R | This study | |

| Reagent/resource | Reference or source | Identifier or catalog number |
|---|---|---|
| *Citrobacter rodentium* Δ*nleB*/*espL* (*espL*), Kan$^R$/Cm$^R$ | This study | |
| *Yersinia pseudotuberculosis* IP32777, Irg$^R$ | Peterson et al, 2017; Zhang and Bliska, 2010 | |
| *E. coli* S17-1λpir | de Lorenzo et al, 1993 | |
| NEB 5-alpha Competent *E. coli* (High Efficiency) | NEB | #C2987H |
| **Recombinant DNA** | | |
| hEBI3 pEF6-V5 | Addgene | #72490; RRID:Addgene_72490 |
| pcDNA3-N-Flag-NLRP3 | Addgene | #75127; RRID:Addgene_75127 |
| pEF6-RIPK3-V5/His | This study | |
| pcDNA-*espL*-200bp-Cm | This study | |
| pKD3 | Addgene | #45604; RRID:Addgene_45604 |
| pKM208 | Addgene | #13077; RRID:Addgene_13077 |
| pGRG36 | Addgene | #16666; RRID:Addgene_16666 |
| pGRG36-*s12-nleB* | This study | |
| pGRG36-*s12-espL* | This study | |
| **Antibodies** | | |
| Caspase-1 (p20) mouse monoclonal antibody | Adipogen | #G-20B-0042, RRID:AB_2490248 |
| Caspase-3 rabbit polyclonal antibody | Cell Signaling Technology | #9662, RRID:AB_331439 |
| Caspase-8 rabbit polyclonal antibody | Cell Signaling Technology | #4927, RRID:AB_2068301 |
| Cleaved caspase-8 (Asp387) rabbit polyclonal antibody | Cell Signaling Technology | #9429, RRID:AB_2068300 |
| Caspase-11 rabbit monoclonal antibody | Abcam | #ab180673, RRID:AB_2923217 |
| GSDMD rabbit monoclonal antibody | Abcam | #ab209845, RRID:AB_2783550 |
| IL-1β goat polyclonal antibody | R&D System | #AF-401-NA, RRID:AB_416684 |
| FLIP rabbit monoclonal antibody | Cell Signaling Technology | #56343; RRID:AB_2799508 |
| RIPK1 rabbit monoclonal antibody | Cell Signaling Technology | #3493; RRID:AB_2305314 |
| RIPK3 rabbit monoclonal antibody | Cell Signaling Technology | #95702; RRID:AB_2721823 |
| p-MLKL (Ser345) rabbit monoclonal antibody | Abcam | #ab196436; RRID:AB_2687465 |
| MLKL rat monoclonal antibody | Abcam | #ab243142; RRID:AB_3083655 |
| NLRP3 mouse monoclonal antibody | Adipogen | #AG-20B-0014; RRID:AB_2490202 |
| MCL-1 rabbit monoclonal antibody | Cell Signaling Technology | #5453; RRID:AB_10694494 |

| Reagent/resource | Reference or source | Identifier or catalog number |
|---|---|---|
| Phospho-IκBα (Ser32/Ser36) mouse monoclonal antibody | Cell Signaling Technology | #9246; RRID:AB_2267145 |
| IκBα rabbit monoclonal antibody | Cell Signaling Technology | #4812; RRID:AB_10694416 |
| p65 rabbit monoclonal antibody | Cell Signaling Technology | #8242; RRID:AB_10859369 |
| p-p65 (Ser536) rabbit monoclonal antibody | Cell Signaling Technology | #3033; RRID:AB_331284 |
| JNK rabbit monoclonal antibody | Cell Signaling Technology | #9258; RRID:AB_2141027 |
| p-JNK (Thr183/Tyr185) rabbit monoclonal antibody | Cell Signaling Technology | #4668; RRID:AB_823588 |
| p38 rabbit polyclonal antibody | Cell Signaling Technology | #9212; RRID:AB_330713 |
| p-p38 (Thr180/Tyr182) rabbit monoclonal antibody | Cell Signaling Technology | #9215; RRID:AB_331762 |
| ERK1/2 rabbit polyclonal antibody | Cell Signaling Technology | #9102; RRID:AB_330744 |
| p-ERK1/2 (Thr202/Tyr204) mouse monoclonal antibody | Cell Signaling Technology | #9106; RRID:AB_331768 |
| IRF3 rabbit monoclonal antibody | Cell Signaling Technology | #4302; RRID:AB_1904036 |
| p-IRF3 (Ser396) rabbit monoclonal antibody | Cell Signaling Technology | #4947, RRID:AB_823547 |
| Alpha-tubulin rabbit polyclonal antibody | Cell Signaling Technology | #2144S, RRID:AB_2210548 |
| GAPDH mouse monoclonal antibody | Abcam | #ab8245; RRID:AB_2107448 |
| Goat anti-Rabbit IgG (H + L) Secondary Antibody, HRP | Invitrogen | #31460; RRID:AB_228341 |
| Donkey anti-Mouse IgG (H + L) Secondary Antibody, HRP | Invitrogen | #A16011; RRID:AB_2534685 |
| Chicken anti-Rat IgG (H + L) Secondary Antibody, HRP | Invitrogen | #A18727; RRID:AB_2535504 |
| Donkey anti-Goat IgG (H + L) Secondary Antibody, HRP | Invitrogen | #A15999; RRID:AB_2534673 |
| InVivoMab anti-mouse TNFα | Bio X Cell | #BE0058; RRID:AB_1107764 |
| InVivoMab rat IgG1 isotype control (anti-HRP) | Bio X Cell | #BE0088; RRID:AB_1107775 |
| **Oligonucleotides and other sequence-based reagents** | | |
| Cloning primer sequences | See Appendix Table S1 | |
| **Chemicals, enzymes and other reagents** | | |
| DMEM, high glucose, no glutamine | ThermoFisher | #11960044 |
| GlutaMAX™ Supplement | ThermoFisher | #35050061 |
| HEPES solution | ThermoFisher | #15630080 |
| MEM Non-Essential Amino Acids Solution (100X) | ThermoFisher | #11140050 |
| Fetal Bovine Serum, qualified, United States | ThermoFisher | #26140079 |

| Reagent/resource | Reference or source | Identifier or catalog number |
| --- | --- | --- |
| Opti-MEM™ I Reduced Serum Medium | ThermoFisher | #31985070 |
| DPBS, no calcium, no magnesium | ThermoFisher | #14190250 |
| 2.5g/l-Trypsin/1 mmol/l-EDTA Solution | Nacalai Tesque | #35554-64 |
| Ultrapure LPS from *E. coli* 055:B55 | Invivogen | #tlrl-pb5lps |
| Pam3CSK4 | Invivogen | #tlrl-pms |
| Recombinant Mouse TNF-α (carrier-free) | BioLegend | #575206 |
| Recombinant Mouse IFN-γ (carrier-free) | BioLegend | #575306 |
| Recombinant Mouse IL-1α (carrier-free) | BioLegend | #575006 |
| Recombinant Mouse IL-1β (carrier-free) | BioLegend | #575102 |
| TPCA | Selleck Chemicals | #S2824 |
| 5Z-7-Oxozeaenol (5z7) | Sigma-Aldrich | #09890 |
| AZD5582 (SMAC) | Selleck Chemicals | #S7362 |
| SM-164 | Selleck Chemicals | #S7089 |
| Nec-1s | Abcam | #ab221984 |
| GSK'872 | Selleck Chemicals | #S8465 |
| QVD-OPh | Abcam | # ab141421 |
| Emricasan | Abcam | #ab287102 |
| MCC950 | Invivogen | #inh-mcc |
| Mouse IL-1 beta/IL-1F2 DuoSet ELISA | R&D System | #DY401 |
| Mouse CXCL2/MIP-2 DuoSet ELISA | R&D System | #DY452 |
| TMB substrate | BD Biosciences | #555214 |
| Incucyte Caspase-3/7 Green Dye | Sartorius | #4440 |
| SYTOX™ Green Nucleic Acid Stain | Invitrogen | #S7020 |
| LDH cytotoxicity detection kit | Sigma-Aldrich | #11644793001 |
| Clarity Western ECL Substrate | Bio-Rad | #1705061 |
| Supersignal® West Femto Maximum Sensitivity Chemiluminescent Substrate | ThermoFisher | #34096 |
| NuPAGE™ LDS Sample Buffer (4X) | Invitrogen | #NP0008 |
| DTT | Bio-Rad | #1610611 |
| NEBuilder® HiFi DNA Assembly Master Mix | NEB | #E2621 |
| Q5® Hot Start High-Fidelity DNA Polymerase | NEB | #M0493S |
| OneTaq Hot Start 2X Master Mix with Standard Buffer | NEB | #M0484S |
| Restriction enzymes | NEB | |
| Nalidixic acid sodium salt | Sigma-Aldrich | #N4382 |

| Reagent/resource | Reference or source | Identifier or catalog number |
| --- | --- | --- |
| Kanamycin monosulfate | Goldbio | #K-120-25 |
| Chloramphenicol | Sigma-Aldrich | #C0378 |
| Ampicillin (sodium) | Sigma-Aldrich | #A-301-25 |
| Irgasan | Sigma-Aldrich | # 72779 |
| Gentamicin | ThermoFisher | #15710072 |
| Blasticidin | Invivogen | #ant-bl-05 |
| Sodium chloride | Sigma-Aldrich | #S9888 |
| Yeast extract | Sigma-Aldrich | # Y1625 |
| Tryptone | Sigma-Aldrich | # T7293 |
| 2x YT medium | Sigma-Aldrich | #Y2377 |
| Magnesium chloride | Sigma-Aldrich | #M8266 |
| Sodium oxalate | Sigma-Aldrich | # 223433 |
| **Software** | | |
| GraphPad Prism 10.0 | Graphpad Software | https://www.graphpad.com |
| Benchling | Benchling | http://www.benchling.com |
| Illustrator | Adobe | https://www.adobe.com |
| Photoshop | Adobe | https://www.adobe.com |
| Incucyte® S3 Software | Sartorius | |
| iBright Analysis Software | ThermoFisher | |
| **Other** | | |
| Biotek Epoch Microplate Reader | Agilent Technologies | |
| iBright™ CL1500 Imaging System | ThermoFisher | |
| Incucyte® S3 Live Cell Analysis System | Sartorius | |

## Mice

C57BL/6J (strain #000664), $Tnfr1^{-/-}$ (strain #003242), $Trif^{Lps2}$ (strain #005037), $Gsdmd^{-/-}$ (strain #032410), $Casp11^{-/-}$ (strain #024698), $Nlrp3^{-/-}$ (strain #021302) and $Mlkl^{SA2}$ (strain #039024) were purchased from Jackson Laboratory and bred in dedicated pathogen-free facilities at the National University of Singapore (NUS). All experiments were performed with the approval from the NUS Institutional Animal Care and Use Committee (approval number R20-1305). Mice between 7-12-week-old of either sex were used for all experiments. Bone marrow from gene-deficient mice were generously provided by Petr Broz ($Ripk3^{-/-}$); James Murphy, James Vince and Kate Lawlor ($Mlkl^{-/-}$ and $Casp8^{-/-}Mlkl^{-/-}$); and Sylvie Alonso ($Ifnar1^{-/-}$).

## Cell culture

Bone marrow-derived macrophages (BMDMs) were differentiated in DMEM supplemented with 20% L929 supernatant (as a source of M-CSF), 10% heat-inactivated FCS, 10 mM HEPES, 1× GlutaMAX and 1× non-essential amino acid (all from Gibco) and used on day

7-9 of differentiation. *Mlkl*⁻/⁻ and *Casp8*⁻/⁻*Mlkl*⁻/⁻ BMDMs were immortalised with Cre-J2 (De Nardo et al, 2018) and immortalised BMDMs (iBMDMs) were maintained in DMEM supplemented with 10% L929 supernatant (as a source of M-CSF), 10% heat-inactivated FCS, 10 mM HEPES, 1× GlutaMAX and 1× non-essential amino acid (all from Gibco). The murine colorectal cell line CMT-93 was purchased from ATCC and maintained in DMEM supplemented with 10% heat-inactivated FCS and 1× GlutaMAX. BMDMs and iBMDMs were routinely seeded at a concentration of $5 \times 10^4$ cells/well in a 96-well plate or $2.5 \times 10^5$ cells/well in a 24-well plate the evening before experiment (BMDMs) or the morning of experiment (iBMDMs). To measure SYTOX Green uptake and caspase-3/7 activity over time in BMDMs, cells were seeded at $1 \times 10^5$ cells/well in a 24-well plate. CMT-93 cells were seeded at $3 \times 10^4$ cells/well in a 96-well plate or $1 \times 10^5$ cells/well in a 24-well plate the evening before experiment. To prime CMT-93 cells for 24 h, cells were seeded at $5 \times 10^4$ cells/well in a 24-well plate the evening before priming.

## Generation of stable CMT-93 cells

Murine RIPK3 (NM_019955) was amplified from BMDM cDNA using Ripk3_F/Ripk3_R, primer pair. RIPK3 PCR product and pEF6-V5/His (Addgene #72490) were digested with NotI and XbaI, and subsequently ligated to generate pEF6-RIPK3-V5/His. The plasmid product was confirmed by sequencing using pEF6-F/pEF6-R. Primer sequences are listed in Appendix Table S1. CMT-93 stably expressing RIPK3 was generated by transfecting pEF6-RIPK3-V5/His using Lipofectamine 2000 (Invitrogen) and selected with 10 μg/ml blasticidin (InvivoGen). After selection, stable CMT-93 cells were maintained with 5 μg/ml blasticidin in routine passaging and 2.5 μg/ml blasticidin upon seeding.

## Bacterial strains and infection assay

*Citrobacter rodentium* ICC169, isogenic Δ*nleB*, Δ*nleE* Δ*nleF* and Δ*espL* (Kelly et al, 2006; Mundy et al, 2003; Pallett et al, 2017; Pearson et al, 2017; Pearson et al, 2013), and *Yersinia pseudotuberculosis* IP32777 (Peterson et al, 2017; Zhang and Bliska, 2010) (see Reagents and Tools table) were previously described. *C. rodentium* was routinely grown in Luria-Bertani (LB) media at 37 °C with aeration overnight, while *Y. pseudotuberculosis* was cultured in 2x YT media at 28 °C overnight with aeration. To induce expression of the type 3 secretion system, overnight *C. rodentium* was diluted 1:75 in warm Opti-MEM and grown statically for 3 h at 37 °C. After which, bacteria were further diluted in Opti-MEM to multiplicity of infection (MOI) of 20 unless stated otherwise. Overnight *Y. pseudotuberculosis* was diluted 1:40 into fresh 2x YT media supplemented with 20 mM magnesium chloride (Sigma-Aldrich) and 20 mM sodium oxalate (Sigma-Aldrich) and grown with aeration at 28 °C for 1 h followed by another 2 h at 37 °C. *Y. pseudotuberculosis* was washed thrice in warm Opti-MEM and diluted to MOI of 20 unless stated otherwise. To prepare stationary phase *C. rodentium* for infection, *C. rodentium* grown at 37 °C overnight was washed thrice in warm Opti-MEM and diluted to MOI of 20. To synchronise infection, plates were centrifuged at 300 $g$ for 10 min at 37 °C. 100 μg/ml of gentamicin (Gibco) were added 1 h post-infection to kill extracellular bacteria.

## Generation of *C. rodentium* Δ*nleB/espL* mutant

To construct Δ*nleB/espL* mutant by lambda Red recombination (Datsenko and Wanner, 2000), the 5' and 3' flanking regions of *espL* were amplified from *C. rodentium* genomic DNA using CRespL-5'HR-F/CRespL-5'HR-R and CRespL-3'HR-F/CRespL-3'HR-R primer pairs. The chloramphenicol resistance cassette was amplified from pKD3 plasmid using CRespL-HR-Cm-F/CRespL-HR-Cm-R primer pair. All three PCR products were ligated in BspDI/XhoI-digested pcDNA (Addgene #75127) using NEBuilder HiFi DNA assembly kit (NEB) to create pcDNA-*espL*-200bp-Cm. The assembled plasmid was confirmed by sequencing using pcDNA-F, pcDNA-R and pcDNA-Cm-F primers. This assembled plasmid was then amplified using EspLHR-F/EspLHR-R primer pair, and the PCR product was electroporated into *C. rodentium* Δ*nleB* expressing pKM208 (Murphy and Campellone, 2003). Subsequently, *C. rodentium* Δ*nleB/espL* transformant was selected on kanamycin/chloramphenicol agar plates at 37 °C overnight and *espL* deletion was confirmed by PCR using EspLHR-F/EspLHR-R primer pair. Primer sequences are listed in Appendix Table S1.

## Cis complementation of *C. rodentium* Δ*nleB/espL*

*C. rodentium* Δ*nleB/espL* was *cis* complemented by Tn7-mediated transgene insertion and complementation strategy (McKenzie and Craig, 2006). To construct transgene insertion vector, *C. rodentium* *s12* promoter was amplified using Cr-S12-F/Cr-S12-R1 or Cr-S12-F/Cr-S12-R2 primer pairs, while *nleB* and *espL* were amplified using S12-Cr-nleB-F/S12-Cr-nleB-R or S12-Cr-espL-F/S12-Cr-espL-R primer pairs respectively from *C. rodentium* genomic DNA. The *s12* promoter and *nleB* or *espL* were ligated in NotI/XhoI-digested pGRG36 (Addgene #16666) using NEBuilder HiFi DNA assembly kit (NEB) to create pGRG36-*s12-nleB* and pGRG36-*s12-espL*. The assembled plasmid was confirmed by sequencing using pGRG36-F, pGRG36-R, Check-Cr-nleB-F2 (for pGRG36-*s12-nleB*) and pGRG36-F, pGRG36-R, Check-Cr-espL-F2 (for pGRG36-*s12-espL*). Either pGRG36 plasmid was introduced into *C. rodentium* Δ*nleB/espL* via conjugation from *Escherichia coli* S17-1λpir donor strain and transgene insertion was performed as previously described (McKenzie and Craig, 2006) to generate *C. rodentium* Δ*nleB/espL* (*nleB*) and Δ*nleB/espL* (*espL*). *nleB* or *espL* insertion at the *att*Tn7 site was confirmed by sequencing using Check-Tn7-F2, Check-Tn7-R, Check-nleB-F2 (for Δ*nleB/espL* (*nleB*)) and Check-Tn7-F2, Check-Tn7-R, Check-espL-F2 primer pair (for Δ*nleB/espL* (*espL*)). Primer sequences are listed in Appendix Table S1.

## LDH assay, caspase-3/7 activity and SYTOX Green uptake

BMDM cytotoxicity was measured using LDH assay (Sigma-Aldrich) according to manufacturer's instructions. LDH release was normalised to 100% lysed cells treated with 0.1% Triton X-100 and expressed as %LDH. Absorbance at 490 nm was measured. To determine caspase-3/7 activity, each well was treated with 1 μM (CMT-93) or 2.5 μM (BMDMs) Incucyte caspase-3/7 green dye (Sartorius), and time-lapse imaging was performed using the Incucyte® S3 Live Cell Analysis System at 20x magnification, 2 to 4

images per well and at 30 min intervals. To measure SYTOX Green uptake in CMT-93 cells, CMT-93 cells and BMDMs were treated with 0.5 μM and 50 nM SYTOX Green (Invitrogen) respectively and imaged at ×20 magnification, 4 images per well and at 30 min intervals. Total green object integrated intensity (GCU × μm²/Image) averaged from wells in duplicates or triplicates was presented as caspase-3/7 activity or SYTOX Green uptake. SYTOX green uptake in BMDMs were normalised to 100% lysed cells treated with 0.1% Triton X-100.

## Detecting NF-κB, MAPK and IRF3 activation by immunoblotting and/or ELISA

CMT-93 cells were infected with *C. rodentium* or isogenic Δ*nleE* for 1 h, after which culture media was removed and primed with 100 ng/ml TNF in fresh Opti-MEM for the indicated time points. Whereas, BMDMs were infected with *C. rodentium* or isogenic Δ*nleE* or primed with 100 ng/ml LPS or 100 ng/ml TNF for the indicated time points. Cell lysates or culture supernatant were harvested for immunoblotting or ELISA, respectively.

## In vitro activation of apoptosis and necroptosis

To activate caspase-8 in vitro, BMDMs were co-treated with 100 ng/ml LPS or 100 ng/ml TNF with either 1 μM AZD5582, 5 μM TPCA or 125 nM 5z7 for 5 h unless stated otherwise. To induce necroptosis in BMDMs, cells were primed with 100 ng/ml LPS for 3 h and stimulated with 10 μM emricasan in the last 30 min of priming followed by 5 h of 5 μM SM-164 stimulation. To induce necroptosis in RIPK3-expressing CMT-93, cells were co-treated with 100 ng/ml TNF and 10 μM emricasan.

## ELISA

Mouse IL-1β ELISA (#DY401, R&D) and CXCL2 ELISA (#DY452, R&D) were carried out according to manufacturer's instructions. TMB substrate (BD Bioscience) was added and absorbance at 450 nm was measured.

## In vivo infection

A single colony of *Citrobacter rodentium* was inoculated into 10 ml of LB and incubated overnight at 37 °C with aeration. On the next day, bacteria were centrifuged at $3000 \times g$ for 10 min at 4 °C and washed thrice with sterile PBS and concentrated to $5 \times 10^9$ colony forming unit (CFU)/ml. 7- to 9-week-old mice were challenged with $1 \times 10^9$ CFU stationary-phase bacteria by oral gavage and monitored at regular intervals. To determine bacterial burden, faeces were collected at the indicated time points, weighed and homogenised in 500 μl sterile PBS. The CFU per g of faeces for every individual mouse was determined by serial dilution on LB agar supplemented with 50 μg/ml nalidixic acid.

## Western blot

Cell-free supernatant was precipitated with methanol and chloroform as previously described (Chan and Chen, 2023) and combined in cell extracts that were lysed with lysis buffer (2% SDS, 66 mM Tris-Cl pH 7.4, 10 mM dithiothreitol, and NuPage LDS sample buffer). Proteins were resolved on 10%, 12% or 15% gels and transferred onto nitrocellulose membranes (Bio-rad) using the Trans-Blot Turbo Transfer System (Bio-rad). Antibodies for immunoblot were against RIPK1 (1:1000, #3493, CST), RIPK3 (1:1000, #95702, CST), MLKL (1:1000, ab243142, Abcam), pMLKL (1:1000, ab196436, Abcam), NLRP3 (1:1000, #AG-20B-0014, Adipogen), GSDMD (1:1000, ab209845, Abcam), pro-IL-1β (1:1000, AF-401-NA, R&D), caspase-1 (1:1000, casper-1, Adipo-Gen), caspase-3 (1:1000, #9662, Cell Signaling), cleaved caspase-8 (1:1000, #9429, Cell Signaling), full length caspase-8 (1:1000, #4927, Cell Signaling), caspase-11 (1:1000, #ab180673, Abcam), FLIP (1:1000, #56343, CST), p-IκBα (1:1000, #9246, CST), IκBα (1:1000, #4812, CST), p-p65 (1:1000, #3033, CST), p65 (1:1000, #8242, CST), p-JNK (1:1000, #4668, CST), JNK (1:1000, #9258, CST), p-p38 (1:1000, #9215, CST), p38 (1:1000, #9212, CST), p-ERK (1:1000, #9106, CST), ERK (1:1000, #9102, CST), p-IRF3 (1:1000, #4947, CST), IRF3 (1:1000, #4302, CST), MCL-1 (1:1000, #5453, CST), GADPH (1:5000, ab8245, Abcam) and tubulin (1:5000; #2144, Cell Signalling). Anti-rabbit, anti-mouse, anti-rat and anti-goat IgG secondary antibodies conjugated to horseradish peroxidase (Invitrogen) were used and developed with Clarity Western Enhanced Chemiluminescence (ECL) Substrate (Bio-rad) or SuperSignal West Femto Maximum Sensitivity Substrate (Thermo Scientific) and imaged using film or the iBright Imager (Invitrogen).

## Statistical analysis

Statistical analyses were performed using Prism 10 (Graphpad) software. Two-way ANOVA with multiple comparisons (Tukey's, Dunnett's or Šídák's) was used for all normally distributed LDH and ELISA datasets with more than two parameters. Unpaired *t* test was used for normally distributed LDH dataset with two parameters. Differences in stool CFU were assessed using Mann–Whitney *t* test, where a normal distribution was not assumed. Data were considered significant when $P \leq 0.05$, with *$P \leq 0.05$, **$P \leq 0.01$, ***$P \leq 0.001$ or ****$P \leq 0.0001$.

# Data availability

This study includes no data deposited in external repositories.

The source data of this paper are collected in the following database record: biostudies:S-SCDT-10_1038-S44318-025-00412-5.

# Peer review information

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

## Acknowledgements

We thank Gad Frankel for providing *Citrobacter rodentium ΔespL*; John Chen for providing pKD3 and pKM208; Yunn-Hwen Gan for providing pGRG36 and *E. coli* S17-1λpir; Petr Broz for providing *Ripk3*$^{-/-}$ bones; Sylvie Alonso for providing *Ifnar1*$^{-/-}$ bones; James Murphy, James Vince and Kate Lawlor for sharing *Mlkl*$^{-/-}$ and *Casp8*$^{-/-}$*Mlkl*$^{-/-}$ BMDMs. HWY is a recipient of the NUS Yong Loo Lin School of Medicine Postgraduate Research Scholarship. This work was supported by grants from the National University Health System (NUHSRO/2020/117/STARTUP/04) and Ministry of Education (MOE-T2EP30123-0006) to KWC.

## Author contributions

**Hui Wen Yeap**: Conceptualization; Data curation; Formal analysis; Validation; Investigation; Visualization; Methodology; Writing—original draft; Writing—review and editing. **Ghin Ray Goh**: Data curation; Formal analysis. **Safwah Nasuha Rosli**: Data curation. **Hai Shin Pung**: Data curation. **Cristina Giogha**: Resources; Writing—review and editing. **Vik Ven Eng**: Resources; Writing—review and editing. **Jaclyn S Pearson**: Resources; Writing—review and editing. **Elizabeth L Hartland**: Resources; Writing—review and editing. **Kaiwen W Chen**: Conceptualization; Data curation; Formal analysis; Supervision; Funding acquisition; Validation; Investigation; Visualization; Methodology; Writing—original draft; Project administration; Writing—review and editing.

Source data underlying figure panels in this paper may have individual authorship assigned. Where available, figure panel/source data authorship is listed in the following database record: biostudies:S-SCDT-10_1038-S44318-025-00412-5.

## Disclosure and competing interests statement

The authors declare no competing interests.

# Expanded View Figures

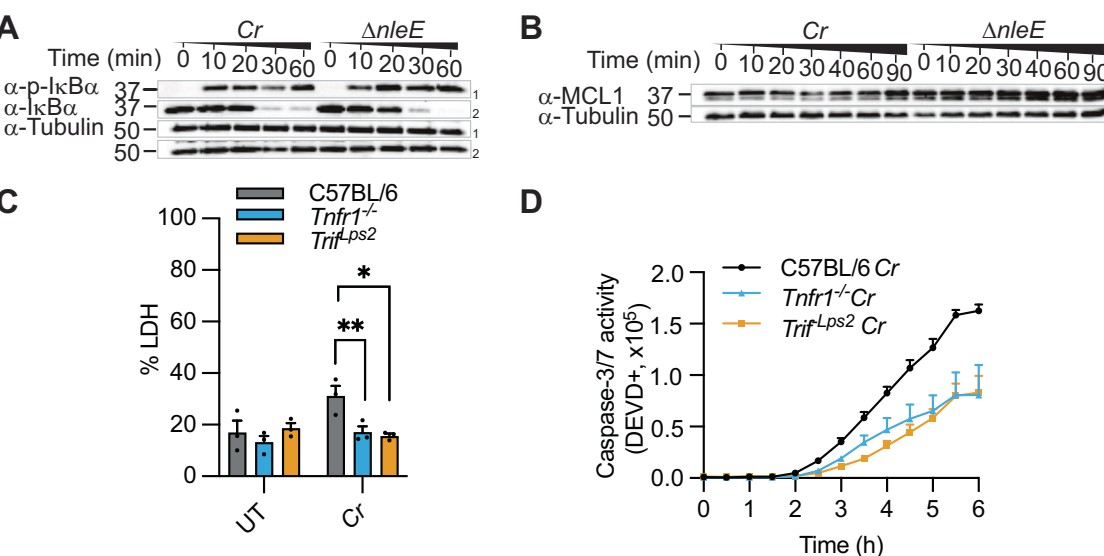

**Figure EV1. NleE marginally suppresses NF-κB signalling in macrophages.**

(A–D) Unprimed BMDMs were challenged with log-phase *C. rodentium* (*Cr*) or Δ*nleE* for (A, B, D) the indicated time points or (C) 3 h. (A, B) Cell lysates were examined by immunoblotting. (C) LDH release was measured. (D) Caspase-3/7 activity (DEVD-positive) was quantified using IncuCyte. (C) Pooled data are mean ± SEM of three independent experiments ($P = 0.0104$ for *Tnfr1*$^{-/-}$ and $P = 0.0056$ for *Trif*$^{Lps2}$). (D) Data are mean ± SD of technical triplicates representative of three independent experiments. All *P* values were calculated with two-way ANOVA test. Data are considered significant when $P \le 0.05$, with *$P \le 0.05$ or **$P \le 0.01$. Source data are available online for this figure.

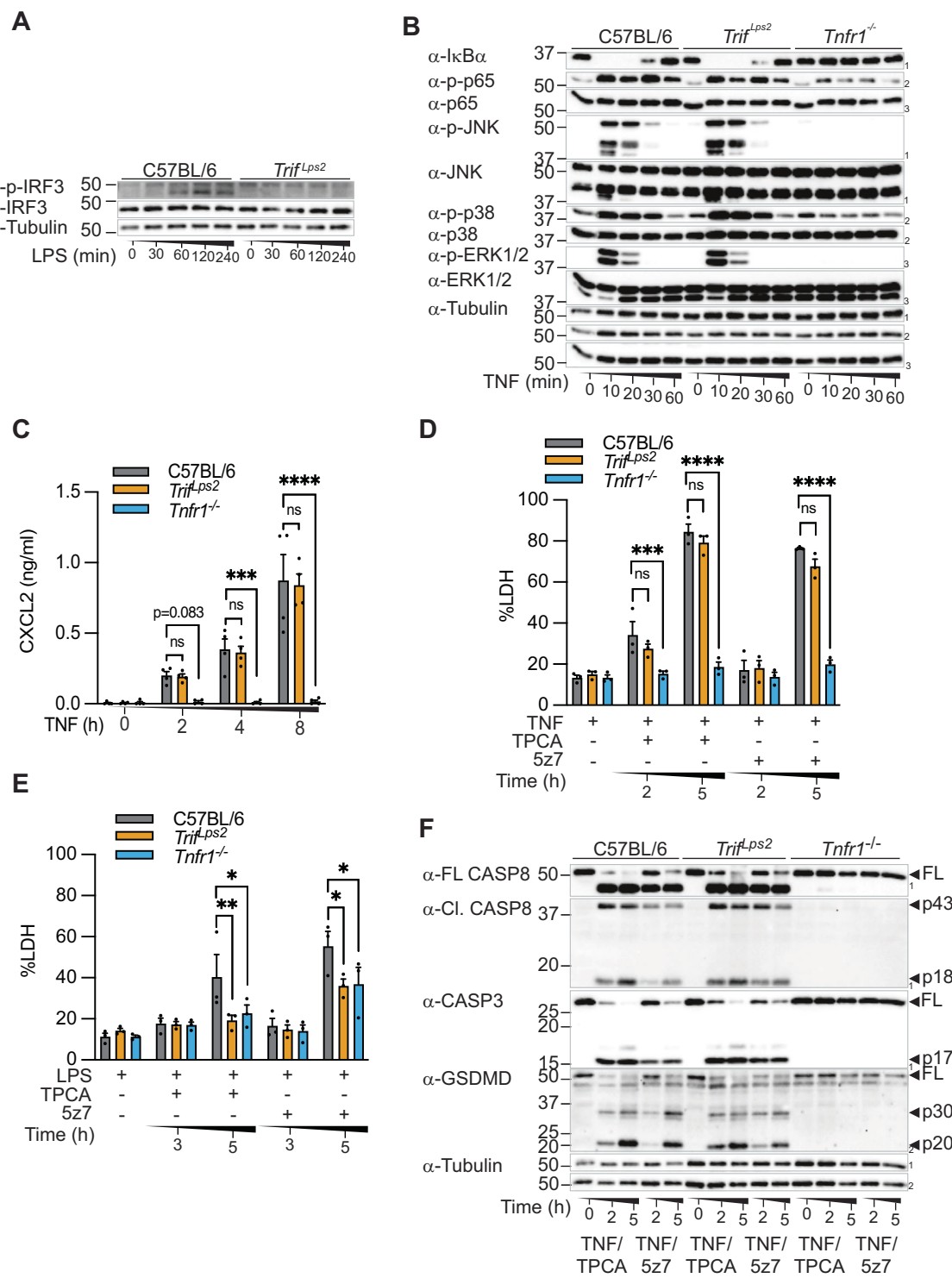

◄ **Figure EV2.  WT and *Trif^Lps2* BMDMs display comparable TNF-mediated inflammation and cell death.**

BMDMs were treated with (**A**) LPS (100 ng/ml) or (**B, C**) TNF (100 ng/ml) for the indicated time points. BMDMs were co-stimulated with (**D, F**) TNF (100 ng/ml) and TPCA (IKK inhibitor; 2.5 μM) or 5z7 (TAK1 inhibitor; 125 nM) or (**E**) LPS (100 ng/ml) and TPCA (2.5 μM) or 5z7 (125 nM) for the indicated time points. (**A, B**) Cell lysates or (**F**) mixed supernatant and lysates were examined by immunoblotting. (**C**) Cytokine secretion and (**D, E**) LDH release were quantified. (**C**) Pooled data are mean ± SEM of four independent experiments ($P = 0.0003$ for 4 h C57BL/6 vs *Tnfr1^−/−^* and $P ≤ 0.0001$ for 8 h C57BL/6 vs *Tnfr1^−/−^*). (**D**) Pooled data are mean ± SEM of three independent experiments ($P = 0.0003$ for 2 h TNF/TPCA treatment in C57BL/6 vs *Tnfr1^−/−^* and $P ≤ 0.0001$ for 5 h TNF/TPCA or TNF/5z7 treatment in C57BL/6 vs *Tnfr1^−/−^*). (**E**) Pooled data are mean ± SEM of three independent experiments (for 5 h LPS/TPCA treatment, $P = 0.0056$ in C57BL/6 vs *Trif^Lps2^* and $P = 0.0208$ in C57BL/6 vs *Tnfr1^−/−^*; for 5 h LPS/5z7 treatment, $P = 0.0112$ in C57BL/6 vs *Trif^Lps2^* and $P = 0.0152$ in C57BL/6 vs *Tnfr1^−/−^*). All P values were calculated with two-way ANOVA test. Data are considered significant when $P ≤ 0.05$, with *$P ≤ 0.05$, **$P ≤ 0.01$, ***$P ≤ 0.001$ or ****$P ≤ 0.0001$. #non-specific band. Source data are available online for this figure.

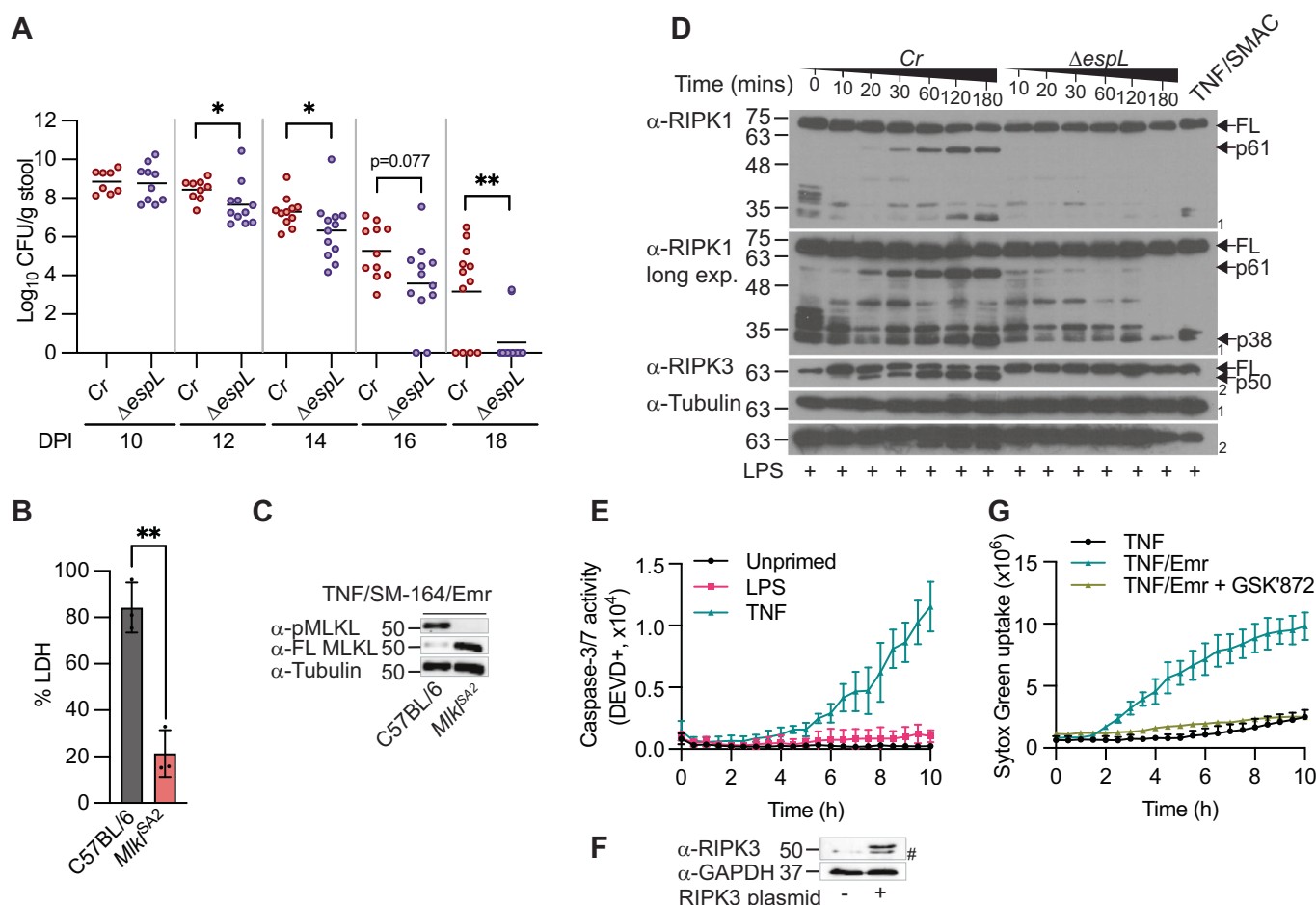

**Figure EV3. EspL is essential for in vivo colonisation and for subverting macrophage necroptosis.**

(A) C57BL/6 mice were infected with *C. rodentium* (Cr) or Δ*espL* and bacterial load in stool was enumerated. Each data point represents log_{10} colony forming unit (CFU) per gram of stool of an individual mouse and means are indicated. Eight to twelve mice were included per group ($P = 0.0251$ for day 12, $P = 0.0225$ for day 14 and $P = 0.0032$ for day 18). (B, C) BMDMs were primed with TNF (100 ng/ml) for 3 h and stimulated with emricasan (10 μM) in the last 30 min of priming followed by 5 h of stimulation with SM-164 (5 μM). (D) BMDMs were primed with LPS (100 ng/ml) for 3 h before infected with log-phase *C. rodentium* (Cr) and Δ*espL* for the indicated time points. (E) CMT-93 were stimulated with LPS (100 ng/ml) or TNF (100 ng/ml) over time. (F) RIPK3 expression in CMT-93 parental cells and CMT-93 stably overexpressing RIPK3 were confirmed by immunoblotting. (G) RIPK3-expressing CMT-93 cells were co-treated with TNF (100 ng/ml) and emricasan (10 μM). (C, D) Mixed supernatant and lysates or (F) cell lysates were examined by immunoblotting. (D) BMDMs co-stimulated with TNF (100 ng/ml)/ SMAC mimetic (1 μM) as positive control showing caspase-8 active cleavage of RIPK1. (B) LDH release were quantified. (E) Caspase-3/7 activity (DEVD-positive) and (G) SYTOX green uptake were quantified using IncuCyte. (B) Pooled data are mean ± SEM of three independent experiments ($P = 0.0018$). Data are mean ± SD of technical (E) triplicates or (G) duplicates representative of three independent experiments. *P* values in (A) and (B) were calculated with Mann–Whitney *t*-test and unpaired *t*-test respectively. Data are considered significant when $P \leq 0.05$, with *$P \leq 0.05$ or **$P \leq 0.01$. Source data are available online for this figure.

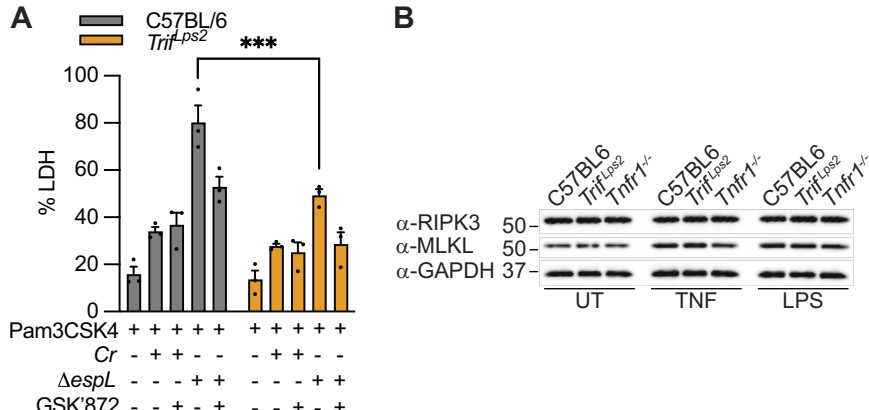

**Figure EV4.** *Trif^{Lps2}* mutation protects Pam3CSK4-primed macrophages from Δ*espL*-induced necroptosis.

(A) BMDMs were primed with Pam3CSK4 (1 µg/ml) for 3 h before infected with log-phase *C. rodentium* (*Cr*) or Δ*espL* for 5 h. LDH release was quantified. Where indicated, cells were treated with GSK'872 (5 µM) for 30 min before infection. (B) BMDMs were stimulated with TNF (100 ng/ml) or LPS (100 ng/ml) for 3 h and cell extracts were analysed by immunoblotting. (A) Pooled data are mean ± SEM of three independent experiments (*P* = 0.0002). *P* value was calculated with two-way ANOVA test. Data are considered significant when *P* ≤ 0.05, with ***P* ≤ 0.001. Source data are available online for this figure.

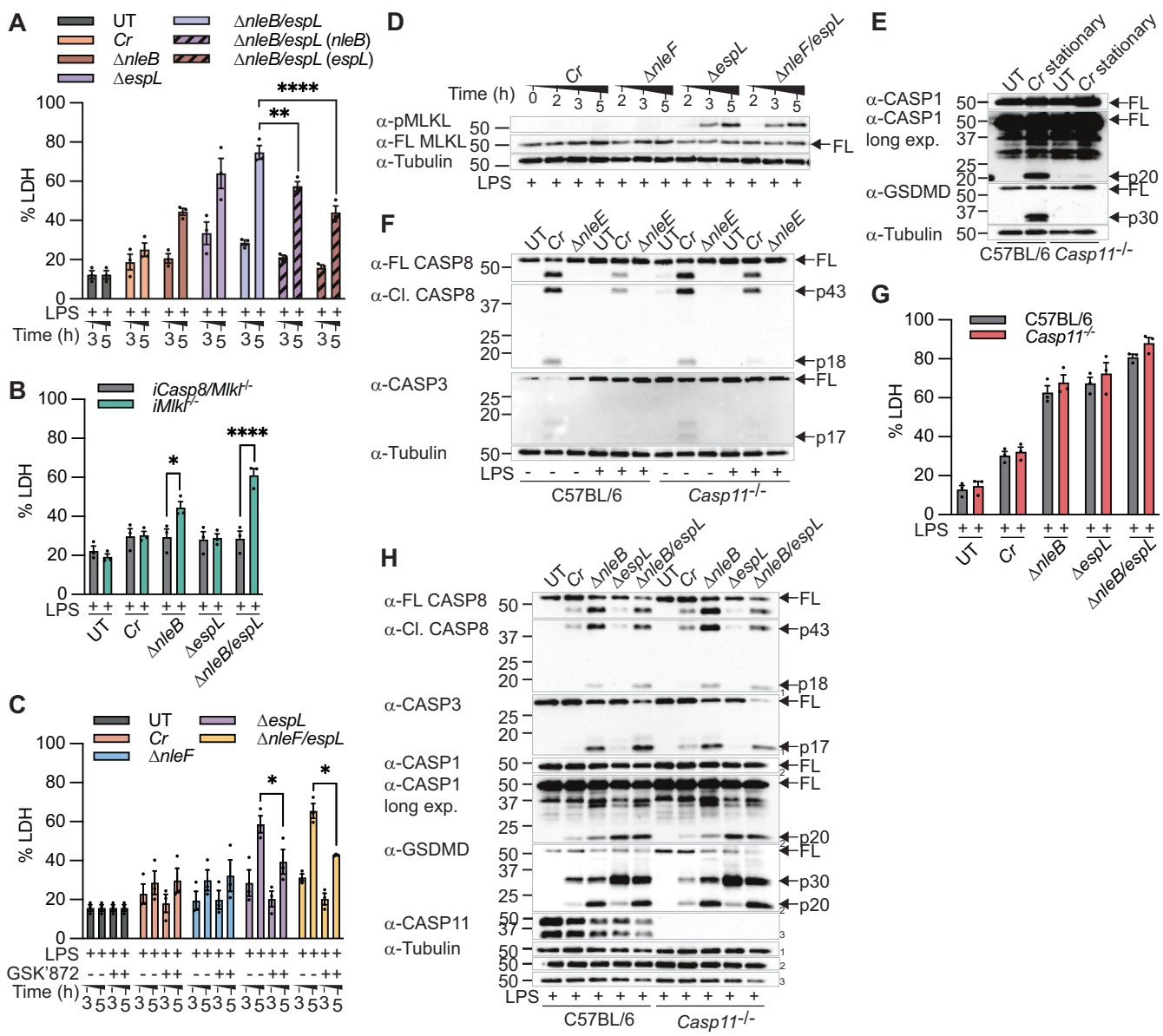

**Figure EV5. Log-phase *C. rodentium* activates caspase-8 but not caspase-11 in macrophages.**

(A–H) BMDMs were unprimed or primed with LPS (100 ng/ml) for 3 h before infected with (A–D, F–H) log- or (E) stationary-phase *C. rodentium* and the various mutants for (A, C, D) the indicated time points, (B, F–H) 5 h or (E) 16 h. (A–C, G) LDH release was quantified. (D–F, H) Mixed supernatant and lysates were examined by immunoblotting. (A) Pooled data are mean ± SEM of three independent experiments ($P = 0.0080$ for Δ*nleB/espL* vs Δ*nleB/espL* (*nleB*) and $P ≤ 0.0001$ for Δ*nleB/espL* vs Δ*nleB/espL* (*espL*)). (B) Pooled data are mean ± SEM of three independent experiments ($P = 0.0153$ for Δ*nleB* and $P ≤ 0.0001$ for Δ*nleB/espL*). (C) Pooled data are mean ± SEM of three independent experiments ($P = 0.0494$ for Δ*espL* treated with GSK'872 and $P = 0.0166$ for Δ*nleF/espL* treated with GSK'872). (G) Pooled data are mean ± SEM of three independent experiments. All $P$ values were calculated with two-way ANOVA test. Data are considered significant when $P ≤ 0.05$, with *$P ≤ 0.05$, **$P ≤ 0.01$, or ****$P ≤ 0.0001$. Source data are available online for this figure.

