## [Peer Review File · The EMBO Journal]

A bacterial network of T3SS effectors counteracts host pro-inflammatory responses and cell death to promote infection

Hui Wen Yeap, Ghin Ray Goh, Safwah Nasuha Rosli, Hai Shin Pung, Cristina Giogha, Vik Ven Eng, Jaclyn Pearson, Elizabeth Hartland, and Kaiwen Chen

Corresponding author(s): Kaiwen Chen (kaiwen.chen@nus.edu.sg)

Review Timeline:

Submission Date:	30th Jul 24
Editorial Decision:	19th Sep 24
Revision Received:	13th Jan 25
Editorial Decision:	19th Feb 25
Revision Received:	21st Feb 25
Accepted:	21st Feb 25

Transaction Report:

The manuscript was originally peer-reviewed in another journal.

Authors' point-by-point reply to Reviewer comments from previous journal

We thank the Reviewers for their time and their constructive feedback. We are delighted that all three Reviewers acknowledged the quality of our work, and Reviewer 3 for highlighting the significance of our study and our efforts in characterising the mechanistic relationship between the microbial effectors in subverting innate signalling and cell death pathways. However, Reviewer 1 and 2 they felt that the novelty of our study is comprised by previous studies that have identified these effectors. We respectfully disagree with their assessment and have listed in detail the novelty of our study below.

Novelty of our study

Reviewer 1 and 2 were less enthusiastic about our manuscript because the three key effectors described in our study, NleE, NleB and EspL were previously identified by our co-author and others. However, these studies did not investigate the mechanistic relationship between these effectors, how they collaborate to subvert innate immune signalling and cell death pathways, and the cellular response to counteract these effectors.

NleE was first reported in 2007 (PMID: 17317262) and that study demonstrated that $\Delta nleE$ are attenuated for *in vivo* colonisation compared to wild type bacteria. Subsequent studies revealed that NleE blocks NF-KB signalling (PMID: 20126447, 20485572) by modifying TAB2/3 (PMID: 22158122), however, none of these studies examined the host response to NleE. Here, we demonstrate that recognition of a single effector, NleE, elicits RIPK1 kinase-dependent apoptosis in macrophages and epithelial cells, and apoptosis caspase activation *in vivo*. We also provide genetic evidence that TNFR1 and TRIF are required for NleE-induced apoptosis during *C. rodentium* infection. We would also like to highlight that while blockade of TAK1/IKK signalling by chemical inhibitors or YopJ from pathogenic *Yersinia* was shown to trigger RIPK1-dependent cell death, this does not necessarily mean that all pathogens that have the capacity to modulate NF-KB signalling will trigger RIPK1-dependent apoptosis by default. For example, *Pseudomonas aeruginosa* encodes C12 that blocks the IKK complex (PMID: 18566250), yet triggers NLR4 inflammasome activation instead of RIPK1-dependent apoptosis (PMID: 18070936). Similarly, *Salmonella Typhimurium* and *Vibrio parahaemolyticus* both encode YopJ homologues (AvrA and VopA respectively), yet they also trigger inflammasome activation instead of RIPK1-dependent apoptosis (PMID: 15190255, 23357873).

NleB was described in 2013 that prevents death receptor signalling by modifying death domain-containing proteins such as FADD and TRADD (PMID: 24025841, 23955153). While both studies revealed a novel post-translational modification (GlcNAcylation) that suppresses FAS and TRAIL signalling, these studies did not examine whether NleB can suppress RIPK1-dependent apoptosis since the field was not aware that NleE triggers RIPK1-dependent apoptosis in the first place (highlighted above). In addition, these studies did not examine the cellular response towards NleB inhibition, which we demonstrate here in our study to be a non-canonical form of necroptosis that requires the scaffolding function of TRIF. We would also like to highlight that while caspase-8 inhibition is known to trigger necroptosis, not all effectors that block caspase-8 have the same potential to trigger necroptosis. For example, we demonstrate in this study that NleF that was previously demonstrated to block caspase-8 activation (PMID: 28138023, 23516580), was unable to sensitise macrophages to necroptosis.

EspL was described more recently in 2017 where our co-authors demonstrate that this microbial protease can cleave RHIM-domain containing proteins such as RIPK1 and RIPK3 (PMID: 28085133), suggesting that EspL can subvert RIPK1 and RIPK3 driven processes such as TNF/TLR3/4-induced proinflammatory signalling, apoptosis and necroptosis. Although the study demonstrated that overexpression of EspL can prevent drug-induced necroptosis (e.g. TNF plus smac mimetic and caspase inhibitor), how and whether *C. rodentium* triggers necroptosis in the first place remains unresolved and remains an important knowledge gap. Here, we demonstrate a complex interplay between the microbial effectors and plasticity of cell death pathways and demonstrate that *C. rodentium* first triggers RIPK1-dependent apoptosis by injecting NleE to subvert TLR4 and TNFR1-dependent

proinflammatory response. To suppress apoptosis, *C. rodentium* then injects a second effector, NleB to block caspase-8 activation, however, this now sensitise macrophages to trigger necroptosis unless EspL is present. We further demonstrate that while NleB and EspL not only collaborates to suppress necroptosis, but also blocks RIPK1-dependent apoptosis and downstream NLRP3 inflammasome activation. Thus, our study provides a major conceptual advancement on the complex interplay between microbial effectors and host cell death pathways.

Reviewer #1

In this manuscript, "A bacterial network of T3SS effectors counteracts host pro-inflammatory responses and cell death to promote infection," Chen and colleagues report the interaction between bacterial T3SS effectors and host cell death pathways. They show that macrophages and colonic epithelial cells activate RIPK1 kinase-dependent caspase-8 activation during *Citrobacter* infection to counteract NleE blockade of NFkB. Furthermore, macrophages undergo RIPK3-mediated necroptosis due to NleB suppression of caspase-8, which is inhibited by a third T3SS effector, EspL. In sum, they delineate the effect of the inhibition of host factors by a complex network of *Citrobacter* effectors on necroptosis. While the results are clear and the data support the conclusions drawn, there is a limited conceptual advancement over what is already known about these bacterial effectors and host cell death pathways.

Main comments.

As the authors stated, it has already been shown that the effectors NleE, NleB, and EspL inhibit NFkB, caspase-8 and necroptosis, respectively. Therefore, the findings are incremental.

Reply 1: Please refer to the paragraph above on 'novelty of our study'.

Fig. 1E. Do CMT-93 cells have a functional TLR4 pathway and respond to LPS? If not, it may explain the observation that LPS priming did not promote caspase-3/7 activity following *C. rodentium* infection in CMT-93 cells.

Reply 2: During the peer review process, we have already generated data that LPS can indeed sensitise CMT-93 cells to promote caspase-3/7 activity, albeit significantly weaker than TNF.

The lack of Ripk3 expression in CMT-93 cells makes them not an ideal cell type to examine the interaction between bacterial effectors and the casp8-necroptotic pathway.

Reply 3: We acknowledge the limitation that CMT-93 cells do not naturally express RIPK3. However, this is the only murine colonic epithelial cell line available and is also routinely used by Gad Frankel, an leading expert on *C. rodentium* pathogenesis (PMID: 30940698, 1610608, 38936619). Other cell lines such as Caco-2 cells are of human origin and is even less suitable for infection with *C. rodentium*, a pathogen that naturally infects mice. In addressing the limitation, we have ectopically expressed RIPK3 in these cells and demonstrate that these cells are now competent in undergoing drug-induced necroptosis.

How does Pam3CSK4 priming sensitise DespL-infected macrophages to necroptotic cell death (Fig. 4F)?

Reply 4: We believe that both LPS and Pam3CSK4 induces the expression of anti-apoptotic factors such as c-FLIP, which works together with NleB to fully suppress caspase-8 and sensitise the cells to necroptosis. To address this question, we will be performing c-FLIP knockdown experiments to

demonstrate that these cells are now no longer sensitive to necroptosis after LPS or Pam3CSK4 stimulation.

Reviewer #2:

The authors propose a complex dance between bacterial effectors and host signaling, that defines *C. rodentium* infection in mice. As one effector acts the other may counteract, steering the immune response through forms of cell death including apoptosis, pyroptosis, and finally necroptosis, centrally regulated by Ripk1, Caspase-8, and downstream Nlrp3. Several useful pieces of information are presented, reporting how *C. rodentium* effectors manipulate the host response to infection at multiple levels. NleE inhibits Tab2/3, NleB inhibits Caspase-8, and EspL inhibits Rhim domain proteins. The depth of the study on some of the mechanisms is however not always satisfying, EspL was the most well characterized. Some signaling events controlled by these effectors are not exactly novel as they have been described in several previous publications, this paper focuses on elucidating pathways of how they together interact to control innate immune signaling. Many of the experiments appear well performed with clear results, although some interpretations could be better supported (see comments below). Ultimately, this work serves as a good source of information about these effectors and contributes towards a model by which this bacterium interferes with cell death signaling. Thus, some of the work may be perceived as less novel but will serve as a foundation for a better understanding of these effectors. *C. rodentium* is a useful rodent model pathogen that shares some aspects of virulence factors (including effector proteins) with more important human-relevant bacteria, but these are not studied in the present manuscript. That may also limit some of the significance of this work.

Specific comments:

1. Figure 1D:

"These suggest that while TNFR1 and TRIF contribute equally to cell rupture, the TRIF pathway dominates apoptotic caspase activation". Trif also dominates GsdmD cleavage according to this blot, so it is unclear if there is a preference for apoptosis specifically, or if it is more dominant in signaling overall vs TNFR. Even though LDH is same at 3hrs, perhaps a timepoint analysis may reveal a difference in rate of cell death in the TRIF KOs. More needs to be done to make the claim of apoptotic preference.

Reply 5: We will perform a time course analysis using WT vs *Tnfr1*^{-/-} vs *Trif*^{-/-} to measure SYTOX Green uptake (membrane permeability) and western blot to demonstrate apoptotic caspase activation.

2. Fig 1H:

It is difficult for the authors to convincingly claim reduced caspase-8 cleavage in nleE infected colon as signal is overall rather low. Caspase-3 reduction is however better presented.

Reply 6: We agree with the Reviewer. Unfortunately despite our best efforts, we are unable to get a clearer caspase-8 blot. Thus, we will reword our text and conclude that caspase-3 activation in the infected colon is NleE-dependent.

3. Fig 2E: "EspL converted full length RIPK3 into the inactive p50 fragment more readily than full length RIPK1 into the p61 fragment (Fig. 2E), suggesting that EspL cleaves RIPK3 more efficiently in mouse macrophages." This seems to be a stretch and over-interpretation of the present data. Comparing different blots like this could be due to antibody sensitivity. EspL appears to negate all Ripk3, but only slightly Ripk1, but since both Ripks are cleaved, could this difference be caused by antibody epitopes and quality? Are there other approaches to demonstrate this claim, and preferably a method that can utilize normalized data? Perhaps a measure of Ripk1 vs Ripk3 activity in the presence of EspL?

Reply 7: We agree with the Reviewer that the sensitivity of the different antibodies may lead to mis-interpretation of our data. To ensure the same epitope sensitivity, we will clone RIPK1 and RIPK3 with a N or C-terminal V5 tag, and determine the cleavage of the tagged proteins after *C. rodentium* infection using an anti-V5 antibody.

4. Fig 4A: "Given that MLKL oligomerization was previously shown to activate NLRP3 inflammasome these findings suggest that TRIF is essential for NLRP3 inflammasome and downstream GSDMD activation during Delta-espL-induced necroptosis (Fig. 4A, Supp. Fig. 4A)." The authors could simply blot for Nlrp3 expression or use Nlrp3 KO cells to directly answer this question. Caspase 1 is affected, but current data does not restrict the possibility of other inflammasomes being responsible.

Reply 7: We will perform these experiments in WT and *Nlrp3*^{-/-} as suggested.

5. 260-263 "Next, to determine whether TRIF promotes macrophage necroptosis by autocrine type I IFN signalling, we challenged LPS-primed WT and *Ifnar1*^{-/-} macrophages with Delta-espL. To our surprise, DespL infection triggered comparable GSK'872-sensitive LDH release in WT and *Ifnar1*^{-/-} It is unclear why the authors find this surprising.

Reply 8: We find that the results were surprising because induction of type I interferon is the best characterised function for TRIF, yet our data shows that necroptosis is unaffected in *Ifnar1*^{-/-} cells. We can reword this sentence easily.

6. 294-298 Surprisingly, we noted that Δ NleB/espL triggered enhanced RIPK1 kinase-dependent caspase-8 activation, GSDMD cleavage and IL-1 β maturation compared to Δ NleB infection in LPS-primed macrophages (Fig. 5D-5E, Supp. Fig. 5C), while Δ NleB/espL complemented with NleB or espL suppressed cell lysis."

It is also here unclear why the authors would find this surprising. Is this not the expected result, based upon the reported actions of the effectors? It would be assumed that deleting NleB sensitized the cells to necroptosis, which produced Nlrp3 dependent IL1 β and GsdmD pores.

Reply 9: Yes, it is expected that removal of NleB will switch the cells back to apoptosis even in the EspL background, since blockade of caspase-8 is a prerequisite for necroptosis. We find this result surprising because we observed that loss of EspL alone has did not significantly enhance apoptosis, yet, removing both NleB and EspL triggered massive apoptosis. This suggest that NleB and EspL not only collaborates to block necroptosis, but also suppresses apoptosis. We can easily reword this to enhance the clarity of our text.

7. 331-336 "Given that RIPK1DD is essential for RIPK1 dimerisation and caspase-8 activation, we reasoned that *C. rodentium* EspL subverts apoptosis by cleaving RIPK1 to liberate the DD. Unexpectedly, we found that loss of espL had minimal impact on the kinetics or magnitude of *C. rodentium*-induced RIPK1-dependent apoptosis in macrophages and colonic epithelial cells. Instead, we found that EspL was required to suppress necroptosis and downstream NLRP3 inflammasome activation and IL-1 β release in TLR1/2/4-primed macrophages, presumably because EspL cleaves RIPK3 more efficiently than RIPK1 in mouse macrophages."

It is clear espL has a significant effect on downstream MLKL and casp1 activation.

Yet, as stated before, there is not enough data to support this Ripk3 preference over Ripk1. There is one blot Fig2E that implies this, but even so, Fig3C actually shows a very slight reduction of Caspase8 cleavage in Δ espL stimulation, often a sign of interference on Ripk1 scaffolding. So Ripk1 interference may still be significant and it is unclear why apoptosis is minimally affected. To claim preference more data must be shown.

Reply 10: This is the same comment as Reply 7. We can easily perform the same experiments listed in Reply 7 to address this. We will also reword the section on caspase-8 processing in Δ espL-infected cells.

8. Some of the experiments rely largely on chemical inhibitors, the conclusions would be better

supported with more data from primary cells deficient in specific signaling molecules (Ripk1, Ripk3, caspase-8)

Reply 10: We can easily address this by repeating most of these experiments with genetic knockouts.

9. The *in vivo* significance of these effectors is interesting, and more data would benefit the paper. For example, the authors mentioned that EspL promotes microbial colonization by disrupting necroptosis of immune cells recruited to the gut. Was there a difference in cell number / necroptotic cell death detected (e.g. Histology or Annexin/7AAD by flow)? Is there a CFU change in Rip3^{-/-} or Mkl1^{-/-} infected with *Citrobacter*? How do the authors explain the discrepancy between BMDMs and CMT cells?

Reply 11: We acknowledge that this is limitation of our study. Although the field has made some progress on detection of necroptosis *in vivo*, these assays are notoriously challenging, in part, due to the lack of specific antibodies and tools (PMID: 33589776, 38750308, 3878309). We have attempted to perform these experiments prior to submission, however, despite our best efforts, we were unable to generate robust, publication-ready data, perhaps because the colonic epithelium has a high turnover rate and the dead cells are constantly released into the faeces. Thus, we hope that Editor and Advisor is sympathetic of this limitation. However, we will be generating conditional knockouts (*Ripk3* or *Mkl1*) to address this in more detail in a follow up study. We would also like to stress that we are very well aware that we do not have these data, and thus have we have only proposed 'subversion of myeloid necroptosis to suppress immune cell recruitment' in the discussion as a potential mechanism.

10. If NleE controls Ripk1-caspase-8 dependent GSDMD activation, does it also control IL-1b and IL-18 release?

Reply 12: We have already done this experiments and indeed NleE also controls IL-1b release in LPS-primed cells.

11. Please draw a model of your pathway.

Reply 13: Sure, we can do this.

Reviewer #3:

In this study, the authors demonstrate that the enteric pathogen *Citrobacter rodentium* has evolved a network of T3SS effectors to block host cell death responses in macrophages. First, the effector NleE triggers caspase 8 activation, likely due to its ability to inhibit pro-inflammatory signaling. In turn, caspase 8 is inhibited by NleB. Inhibition of caspase 8 triggers necroptosis, which is also blocked by another effector EspL. Therefore, *C. rodentium* has many strategies to inhibit immune cell responses. While the functions of NleE, NleB, and EspL have been described before by some of these authors (PMIDs: 28085133, 24025841) the manuscript delves further into the mechanistic relationship between these effectors, and makes a potentially important contribution to our overall understanding of the field. The manuscript is clear, well-written, and the experiments well done. The studies fit into an overall framework of host-pathogen manipulation of cell death signaling in which the host detects one virulence factor while another then interferes with the second layer of detection. The manuscript represents a potential advance in our understanding of host-pathogen biology, but several conclusions are currently overstated and should be appropriately revised or experimentally addressed to make the stated conclusion. Most of the studies are performed in macrophages; however, the native habitat for *Cr* is thought to be the intestinal epithelium surface. This should be at least discussed or experimentally addressed in places where there is no corroborating evidence for parallel signaling pathways between macrophages and colonic epithelial cells. Additionally, the *in vivo* work

with EspL mutants should be extended as it currently recapitulates previously published data in the field and does not formally demonstrate that EspL manipulates necroptosis in vivo in order to maintain infection.

Major concerns:

1. In figures 1A and 1B, it is clear that RIPK1 is required for Cr-induced cell death and caspase 8 activation. However, it is unclear whether RIPK1 is driving cell death via caspase-8 or MLKL. The authors should perform LDH in their caspase 8^{-/-}mlkl^{-/-} and mlkl^{-/-} BMDMs to formally show that cell death is caspase 8-dependent. This would strengthen their conclusion that *C. rodentium* induces RIPK1-dependent caspase 8-induced cell death.

Reply 13: Sure, we can easily address this by performing LDH release assay in *Casp8^{-/-}Mkl1^{-/-}* and *Mkl1^{-/-}* cells

2. In figure 1C, the authors are looking at the contribution of TNFR1 and TRIF to the activation of cell death. In figure 1C, the authors state that "Tnfr1 and Trif deficiency partially protected macrophages from cell lysis compared to WT cells". Genetic deletion of either of these proteins leads to a decrease in LDH comparable to uninfected cells which suggests that these proteins completely protect macrophages at this timepoint. However, the levels of LDH at the timepoint used for figure 1C (3hr) is only slightly above the level of uninfected cells, which makes it hard to disentangle the individual contribution of TNFR1 and TRIF to cell death. In Figure 1A, the authors saw higher levels of cell death at 5hr post-infection compared to 3hr in figure 1C, so it is recommended that the authors look at LDH at a later timepoint to assess the individual roles of TNFR1 and TRIF to cell death.

Reply 14: Sure, we can easily address this by performing experiments described in Reply 5.

3. In Figure 1D, the authors are looking at the contribution of TNFR1 and TRIF to the cleavage of cell death proteins. In the Tnfr1^{-/-} BMDMs, it is hard to see any decrease in apoptotic caspase cleavage, though there is a reduction in GSDMD cleavage. Therefore, it is hard to draw any conclusions about the contribution of TNFR1 to caspase cleavage. In the Trif^{-/-} BMDMs, there is a clear decrease in caspase cleavage although it is not complete. Presumably, the remaining cleavage is due to TNFR1. In order to strengthen the conclusion that TNFR1 is contributing to caspase cleavage, the authors should perform these experiments in Tnfr1^{-/-}Trif^{-/-} double knockout BMDMs. The expectation would be that if Tnfr1 is contributing to the remaining caspase cleavage in the Trif^{-/-} that there would be a complete abrogation of caspase cleavage in the Tnfr1^{-/-}Trif^{-/-} BMDMs. This would strengthen the conclusion that TNFR1 has a role in promoting apoptotic caspase cleavage, even though Trif dominates caspase activation.

Reply 15: We have already done this experiments during peer review where we supplemented anti-TNF neutralising antibodies onto *Trif^{-/-}* cells to completely abrogate TNFR1 and TRIF signalling.

4. Related to this - Muendlein et al (Poltorak lab) have several sets of studies on the TRIFosome in regulating activation of Casp8-dependent cell death. Minimally these studies should be cited and discussed in the context of this work. Of note, one study (PMID 36563168) highlights a key role for TRIF in driving TNF-induced cytotoxicity and lethal inflammation - this seems particularly relevant and potentially mechanistically related. In this particular setting, is TRIF playing a role downstream of TNFR1, similar to that described by Muendlein et al, or downstream of TLR4 signaling?

Reply 16: We thank the Reviewer for highlighting the study from Muendlein et al. (PMID: 36563168). In fact, we have done a number of experiments on this TNFR1-TRIF axis, but we left the data out on purpose as we and several collaborators (personal communications) were unable to reproduce these data where we found no role for TRIF in TNF-induced NF-KB, MAPK kinase, apoptosis and necroptosis. To address this concern, and to improve scientific reproducibility, we will be happy to include these data in our manuscript.

5. For figure 2D, it is challenging to interpret the caspase 3/7+ cells from the 3-5hr timepoints shown in the images. Quantification of this data is recommended for better visualization and to strengthen the conclusions.

Reply 17: Sure, we can do this.

6. For Fig 1, the authors conclude that "In summary, we uncovered caspase-8-mediated cell death in macrophages, epithelial cells, and infected colon as a host response to counteract NleE subversion of NF-kB activation." The title of figure 1 is also: NleE inhibition of TAB2/3 signalling sensitises macrophages and colonic epithelial cells to RIPK1-dependent caspase-8 activation. This appears to be an overstatement or mis-statement, as, this reviewer did not find any data in the manuscript that demonstrates that NleE inhibits TAB2/3 in BMDMs or the colonic epithelial cell line used in this study. Do the authors have a catalytic mutant that specifically inactivates the TAB2/3 inhibition? In the absence of a direct test, it is not clear that the conclusion that caspase 8 is activated "to counteract NleE subversion of NF-kB activation" can be drawn. This may be the model given all the data in other cell lines such as HeLa and HEK cells but this has not been formally tested.

Reply 18: Sure, we can address this by performing immunoblots for NF-KB and also cytokine release assays following infection with *C. rodentium* or $\Delta nleE$ mutant.

7. In figure 5E, the authors use the Nlrp3 inhibitor MCC950 and they see decreased GSDMD and IL-1 β processing. They conclude that "RIPK1 kinase-dependent caspase-8 activation promotes NLRP3 inflammasome activation upon $\Delta nleB/espL$ infection". However, they should use caspase 8-deficient BMDMs to formally demonstrate this. Caspase 11 has also been shown to activate the Nlrp3 inflammasome during *Citrobacter* infection (PMID: 22819539), so it is possible that caspase-11 drives Nlrp3 activation and not caspase 8. The authors should infect Caspase 8 $^{-/-}$ BMDMs as well as Casp11 $^{-/-}$ BMDMs and assess IL-1 β and Gsdmd processing as well as ASC speck formation to formally show that caspase 8 promotes Nlrp3 inflammasome activation.

Reply 19: Sure, we can address this by performing the experiments by comparing caspase-1, GSDMD and IL-1 processing in WT versus *Casp8 $^{-/-}$* and *Casp11 $^{-/-}$* macrophages.

8. The conclusion that EspL is essential for intestinal colonization is a bit too strong and probably incorrect, given no difference 10 days post-infection (Fig. 3A). There is more rapid clearance of the mutant, so perhaps persistence or maintenance of infection is affected but certainly not a difference in colonization per se. This conclusion should be modified. Indeed, some of the authors previously demonstrated that the EPEC EspL homolog cleaves and degrades RHIM domain proteins, and also demonstrated that EspL contributes to infection maintenance (PMID 28085133). This data therefore recapitulates previously published work and would need to be extended in order to be a substantive advance in our understanding of the biology. Minimally, it would be critical for the authors to formally demonstrate that the effect of espL on in vivo infection is due to its effect on necroptosis (which has not been done to my knowledge). While this would not necessarily be a major conceptual advance over what we currently think in the field, it would nonetheless be important since EspL may have other functions in vivo. The authors should perform the infections in RIPK3 (or MLKL) as well as RIPK1-kinase deficient mice to test whether the effect of EspL on necroptosis is important for the effect they observe on in vivo infection. Furthermore, the authors should perform histopathology on infected tissues in WT and RIPK3 $^{-/-}$ mice as well as immuno-stain for phospho-RIPK3 in tissues in response to wt and EspL mutant infection.

Reply 20: We agree with the Reviewer that the term 'colonisation' is incorrect, and we will replace it with 'persistence' or 'maintenance'. We also agree that comparing bacterial load in WT versus *Mik1 $^{-/-}$* mice following $\Delta espL$ infection will provide additional confirmation that EspL indeed subverts necroptosis *in vivo*. Therefore, we have ordered *Mik1*-deficient animals from JAX in January this year,

however, the colony has is still not ready and we do not have access to these animals. In addition, our vivarium for *C. rodentium* work is also undergoing renovation for the next two to three months, therefore, given these technical difficulties, we are not confident that we will be able to address this concern in a timely manner and we hope that our existing data are sufficient for publication in EMBOJ. We are planning to address this issue as part of our future conditional knockout studies (see also Reply 11). Please also refer to Reply 11 regarding immunostaining of infected tissues.

9. More broadly - most of the in vitro and mechanistic studies are performed in macrophages, with a few corroborating studies in a colonic epithelial cell line. However, the natural habitat for EPEC and *Citrobacter* is at the surface of intestinal epithelial cells. How do the findings in macrophages, particularly in figures 4 and 5 translate to epithelial cells? It is now becoming well appreciated that there are cell type-specific differences in the signaling pathways and outcomes of pyroptosis/apoptosis/necroptosis cell death in different cell types, so it seems to me that this is an important aspect of the study that should be looked at.

Reply 21: We agree with the Reviewer that there are cell-type specific differences, thus, we have thoroughly examined both apoptosis and necroptosis in macrophages and epithelial cells. Our data revealed that NleE triggers RIPK1-dependent apoptosis in both macrophages and epithelial cells, which is suppressed by NleB in both cell types. However, EspL only seem to subvert necroptosis in macrophages but not epithelial cells, even when RIPK3 was overexpressed. We now have additional data that similar to macrophages, NleB and EspL collaborates to block apoptosis in epithelial cells. Thus, we believe that we have addressed to the best of our ability with the current available tools, cell type specific responses between macrophages and epithelial cells. We will continue to address this *in vivo* by as part of our future conditional knockout studies (see also Reply 11).

Minor concerns:

1. The NleB/EspL interaction in which a virulence factor blocks caspase-8 activation and potentially unleashes necroptosis, which needs to be blocked by another virulence factor is highly reminiscent of the vIRA/vICA proteins in hCMV (see studies by Kaiser/Upton/Mocarski et al). this work should be cited and discussed.

Reply 21: Sure, we can cite this study.

2. Some of the co-authors have previously demonstrated that NleB1 from EPEC directly binds and modifies FADD/TRADD/RIPK1 to prevent activation of the DISC or Complex IIa. This is the presumed function of NleB in *Citrobacter* here. Is there direct evidence for this?

Reply 22: The ability for NleB from *Citrobacter rodentium* to modify FADD was already demonstrated previously (PMCID: PMC5655511).

3. For supporting figure 4B, it is confusing that the authors see decreased LDH release in the Trif-/- BMDMs infected with the espL mutant, given that they stimulated with the TLR1/2 agonist Pam3CSK4, which is TRIF-independent. This suggests that TRIF is playing a role independent of the TLR being stimulated - potentially downstream of TNFR1 as per Muendlein. Although the authors address this in the discussion section, they should also mention it in the results text for clarity to the reader.

Reply 23: Sure, we can discuss this in more detail in relation to Muendlein et al. See also Reply 16.

Dear Kaiwen,

Thank you again for transferring to The EMBO Journal for our consideration your manuscript (EMBOJ-2024-118621) along with your rebuttal letter to the comments of the reviewers who previously assessed it at another journal, as well as your provisional revision plan. We have shared your manuscript and your point-by-point response with two additional arbitrators with familiarity both with the field and our journal and its bar, and we have now received their comments, which I have already shared with you (they are included again below). Thank you again for your patience during this rather protracted process, and for your response to our arbitrators' comments, which was very helpful for us to reach a fair and balanced decision on your manuscript.

I am glad to say that our arbitrators are generally supportive of the study and explain that it provides a comprehensive dissection of the complex host-pathogen interactions involving *Citrobacter rodentium*. They find your suggested revision plan sufficient to significantly improve the quality of the manuscript by addressing the majority of the concerns raised by the original referees. They also provide a few additional suggestions for strengthening the manuscript further.

In your response to these comments you expressed your willingness to address the concerns, with the exception of the experiments using littermate controls and Mkl^{-/-} mice (pointed out by arbitrator 2), due to practical limitations. Although we agree with the original referees and the advisor that these data would further strengthen your study and provide additional support to some of your conclusions, these data will not be required for further consideration of your manuscript at The EMBO Journal.

In light of the favorable input we received from our arbitrators, and the extensive revision you have outlined in your rebuttal letter and your response to our arbitrators, I would like to invite you to submit a revised version of the manuscript -according to your provisional plan- along with a detailed point-by-point response addressing the referees' and arbitrators' comments and describing all changes and additions to the manuscript. I should add that it is The EMBO Journal policy to allow a single round of major revision, and acceptance of your manuscript will therefore depend on the completeness of your responses in this revised version. Please let me know if you have any questions or comments that you would like to discuss with me.

We generally allow three months as standard revision time (December 18, 2024), but we may be able to grant an extension to allow enough time for the revision should the need arise. Should you foresee a problem in meeting the three-month deadline, please let us know. As a matter of policy, competing manuscripts published during this period will not negatively impact our assessment of the conceptual advance presented by your study. However, we request that you contact us as soon as possible upon publication of any related work, to discuss how to proceed.

Thank you again for the opportunity to consider your work for publication in The EMBO Journal. I am looking forward to your revision.

Best regards,

Ioannis

Instructions for preparing your revised manuscript

1. When you are ready to submit the revision, please upload:

- A Word file of the manuscript text (including legends of main Figures, EV Figures and Tables). Please make sure that changes are highlighted (or "tracked") to be clearly visible.

- Individual production-quality figure files (one file per figure). When assembling your figures, please refer to our figure preparation guidelines in order to ensure proper formatting and readability in print as well as on screen:

If the data shown in a figure are obtained from n {less than or equal to} 2, please use scatter plots showing the individual data points.

- i. the name of the statistical test used to generate error bars and P values
- ii. the number (n) of independent experiments (please specify technical or biological replicates) underlying each data point (discussion of statistical methodology can be reported in the Materials and Methods section, but figure legends should contain a basic description of n, P, and the test applied)
- iii. the nature of the bars and error bars (s.d., s.e.m.).

- A point-by-point response to the referees' comments, with a detailed description of the changes made (as a word file). All referees' concerns must be fully addressed and their suggestions taken on board. When preparing your letter of response to the referees' comments, please bear in mind that this will form part of the Review Process File and will therefore be available online to the community. Please note that you have the possibility to opt out of the transparent process at any stage prior to publication by letting the editorial office know (contact@embojournal.org); if you do opt out, the Review Process File link will point to the following statement: "No Peer Review File is available with this article, as the authors have chosen not to make the review process public in this case.". For more details on our Transparent Editorial Process, please visit our website: <https://www.embopress.org/page/journal/14602075/authorguide#transparentprocess>

- Expanded View (EV) files (replacing Supplementary Information) that are collapsible/expandable online. A maximum of 5 EV Figures can be typeset. EV Figures should be cited as "Figure EV1, Figure EV2" etc. in the text, and their respective legends should be included in the manuscript file after the legends of regular figures. See detailed instructions regarding Expanded View files here:

- For the figures that you do NOT wish to display as Expanded View figures, they should be bundled together with their legends in a single PDF file called "Appendix", which should start with a short Table of Contents (including page numbers). Appendix figures should be referred to in the main text as: "Appendix Figure S1, Appendix Figure S2" etc. Please see detailed instructions here: <https://www.embopress.org/page/journal/14602075/authorguide#expandedview>

- A complete author checklist, which you can download from our author guidelines (<https://www.embopress.org/page/journal/14602075/authorguide>). Please note that the checklist will also be part of the Review Process File.

2. Please note that no statistics should be calculated and shown in Figures if n=2. Please also note that each p value should be reported as an exact value.

3. Before submitting your revision, primary datasets (and computer code, where appropriate) produced in this study need to be deposited in appropriate public databases (see <https://www.embopress.org/page/journal/14602075/authorguide#dataavailability>). Their accession numbers, databases, and the specific URLs (links) should be listed in a formal "Data availability" section (placed after Methods).

*** The Data Availability Section is restricted to new primary data that are part of this study. In case you have no data that require deposition in a public database, please state so instead of referring to the database: "Our study includes no data deposited in public repositories." under the heading "Data availability". ***

*** All links should resolve to a page where the data can be accessed. ***

*** Please remember to provide in the Data availability section of your revised manuscript reviewer passwords if the datasets are not yet public. ***

*** Please use detailed data citations for already available datasets that were re-analyzed in your study - for more information on the format, see point #9 below. ***

4. Please check that the title and the abstract of the manuscript are brief, yet explicit, even to non-specialists. The length of the title should not exceed 100 characters, and the abstract should be a single paragraph not exceeding 175 words.

5. All materials and methods need to be described in the manuscript using our "Structured Methods" format, which is now required for all research articles. According to this format, the Methods section includes a single "Reagents and Tools Table" - listing key reagents, experimental models, software and relevant equipment including their sources and relevant identifiers- followed by a "Methods and Protocols" section describing the methods. Please download and fill our Reagents and Tools Table template (.docx), which you can find in our author guide:

<https://www.embopress.org/page/journal/14602075/authorguide#structuredmethods>. When submitting your revised manuscript, please do not include the Reagents and Tools Table in the Methods section of the manuscript but upload it as a separate file choosing the file type "Reagent Table".

6. Please also note our reference format: <https://www.embopress.org/page/journal/14602075/authorguide#referencesformat>.

7. At EMBO Press we ask authors to provide source data for the main manuscript figures. Our source data coordinator will contact you to discuss which figure panels we would need source data for and will also provide you with helpful tips on how to upload and organize the files.
8. Please remember: digital image enhancement is acceptable practice, as long as it accurately represents the original data and conforms to community standards. If a figure has been subjected to significant electronic manipulation, this must be noted in the figure legend or in the "Materials and Methods" section. The editors reserve the right to request original versions of figures and the original images that were used to assemble the figure.
9. Our journal encourages inclusion of data citations in the reference list to directly cite datasets that were obtained from public databases. Data citations in the article text are distinct from normal bibliographical citations and should directly link to the database records from which the data can be accessed. In the main text, data citations are formatted as follows: "Data ref: Smith et al, 2001" or "Data ref: NCBI Sequence Read Archive PRJNA342805, 2017". In the Reference list, data citations must be labeled with "[DATASET]". A data reference must provide the database name, accession number/identifiers, and a resolvable link to the landing page from which the data can be accessed at the end of the reference. Further instructions are available at: <https://www.embopress.org/page/journal/14602075/authorguide#referencesformat>.
10. We request authors to consider both actual and perceived competing interests. Please review our policy (<https://www.embopress.org/page/journal/14602075/authorguide#conflictsofinterest>) and update your competing interests statement if necessary. Please name this section 'Disclosure and competing interests statement' and place it after the Acknowledgements section.
11. Please note that all corresponding authors are required to provide an ORCID ID upon submission of a revised manuscript (<https://orcid.org/>). Please find instructions on how to link your ORCID ID to your account in our manuscript tracking system in our Author guidelines (<https://www.embopress.org/page/journal/14602075/authorguide#authorshipguidelines>).
12. We use CRediT to specify the contributions of each author in the journal submission system. CRediT replaces the author contribution section, which should be removed from the manuscript. Please use the free text box to provide more detailed descriptions. See also guide to authors: <https://www.embopress.org/page/journal/14602075/authorguide#authorshipguidelines>.
13. Further information is available in our Guide For Authors: <https://www.embopress.org/page/journal/14602075/authorguide>
14. We would also welcome the submission of cover suggestions or motifs to be used by our Graphics Illustrator in designing a cover.
15. Please use the link below to submit your revision:
<https://emboj.msubmit.net/cgi-bin/main.plex>

Arbitrator #1:

Here, the authors provide an elegant and comprehensive dissection of host:pathogen interactions, caused by *C. rodentium*, a natural murine pathogen. They demonstrate that infection of macrophages and enterocytes by *C. rodentium* results in the inhibition of Tak1 by NleE to block activation of innate immune signalling. This is sensed by RIPK1, causing its lethal activation. RIPK1 mediated cell death is thereby blocked by two other *C. rodentium*-encoded effectors, NleB and EspL, which interfere with caspase-8 activation (apoptosis) and RIPK3 activation (necroptosis). RIPK1 can also activate inflammasomes if apoptosis or necroptosis is blocked, yet, *C. rodentium* also blocks this cell death modality via the coordinated action of NleB and EspL.

Although earlier studies have biochemically and functionally elucidated individual *C. rodentium* T3SS effector functions, the interplay between the bacterial effectors and how the host counteracts these bacterial effectors was only incompletely shown. Here, the authors integrate the various aspects into a coherent host:pathogen interplay, demonstrating how pathogens and hosts counteract each others weaponry.

All in all, I think they have addressed all the points of the previous reviewers. I have only a few minor points to add, some of it the authors might want to test experimentally to corroborate their findings and ensure that their cellular system is not flawed due to their use of conditioned medium (that also contains a lot of TNF) of L929 as a source of G-CSF.

Minor point:

My only suggestion is to streamline the story somewhat making it more accessible to a broader readership.

Of note, the cell line MC38-K is a colorectal carcinoma that expresses good amounts of RIPK3 and readily dies by necroptosis.

Tone down the conclusion, stating: 'these processes required RIPK1 kinase activity since LDH release or apoptotic caspase activation were significantly reduced in the presence of the RIPK1 kinase inhibitor, Nec-1s'. The authors should be aware that Nec-1s also blocks processes that does not require RIPK1, such as TRIF-mediated or ZBP-induced necroptosis. Nec-1s turns RIPK1 into a dominant-negative molecule through a poorly understood mechanism.

Is it possible that TRIF induced IRF3 activation is sufficient to drive Mlkl induction? This could easily be tested, or alternatively, the authors may amend their ms to suggest this possibility.

Trif^{-/-} macrophages generally have a fundamentally different baseline level of interferon target genes, such as MLKL. Can the authors evaluate how comparable the baseline levels of Mlkl (and other interferon target genes such as ISG15) are between WT, Tnfr1^{-/-} and Trif^{-/-}. If indeed the baseline levels are very different, then it is difficult to conclude that the observed effect is indeed due to TRIF signalling, rather than TNFR1. It may simply reflect the different baseline levels of MLKL. It would be worthwhile to evaluate this.

Along a similar line, I am surprised to see that their macrophages are more prone to undergo apoptosis. In our hands, macrophages (differentiated with G-CSF) are prone to undergo necroptosis. I have noticed that the authors use conditioned medium from L929 cells as a source of G-CSF. However, the authors should be aware that L929 produce a high amount of TNF in addition to G-CSF. Hence, macrophages that are sensitive to TNF will be killed and only resistant cells will emerge. This may skew their findings.

Arbitrator #2:

The manuscript was previously submitted to another journal where it was reviewed and rejected after review. Despite rejection, all three reviewers showed some interest in the study and provided the authors with comments/suggestions. In their point-by-point response, the authors defend the novelty of their findings and describe the editorial changes and new experiments that they are planning to do in the hope for a revision at the EMBO Journal.

While the revision plan of the authors will improve the overall quality of the manuscript, it will not address the concern on the physiological importance of the cell death modalities induced in response to the T3SS effectors for the host immune response against the bacterial infection. As suggested by reviewers 2&3, it seems important to additionally include in vivo infection experiments (including mutant *Citrobacter* strains) in RIPK3 KO (or MLKL KO) and RIPK1 KD mice to demonstrate the effect of the effector(s) on the cell death modality, and the importance of its induction for the host immune response.

Additional remarks:

-Any KO cell line should be studied in comparison to the proper control WT cell line, and not just any WT cells. For instance, Tnfr1^{-/-} and Trif^{-/-} BMDMs should respectively be compared to Tnfr^{+/+} and Trif^{+/+} BMDMs isolated from littermates, and not from commercial C57BL/6 mice.

-Immunoblot results from different genotypes can only be compared if run on the same gel. This is obviously not the case in Figure 4A.

-Line 168: "Host response to counteract NleE subversion of NF-κB activation". However, NF-κB inhibition was previously demonstrated not to lead to RIPK1 activation, in contrast to IKK inhibition (see PMID: 37357102). The authors should better define and demonstrate how NleE induces RIPK1 kinase-dependent (and -independent) apoptosis. Blockade of TAB2/3 is not shown. As the authors make the comparison with *Y. pseudotuberculosis* (which directly target TAK1/IKK), it would be important to evaluate the effect of NleE on TAK1 and IKK.

-Legend of Figure 4A indicates "Mixed supernatant and lysates were analyzed by immunoblotting". The analysis of the cell lysates and supernatant should not be mixed but analyzed separately.

We thank the Advisors for their time and the constructive feedback on our manuscript. Our revised manuscript now includes 34 new figure panels and text changes to improve the conclusion and clarity of our work. We strongly believe that the revised manuscript is now a strong candidate for publication at *EMBOJ*.

Point-by-point reply to Advisor comments

Arbitrator 1

Here, the authors provide an elegant and comprehensive dissection of host: pathogen interactions, caused by *C. rodentium*, a natural murine pathogen. They demonstrate that infection of macrophages and enterocytes by *C. rodentium* results in the inhibition of Tak1 by NleE to block activation of innate immune signalling. This is sensed by RIPK1, causing its lethal activation. RIPK1 mediated cell death is thereby blocked by two other *C. rodentium*-encoded effectors, NleB and EspL, which interfere with caspase-8 activation (apoptosis) and RIPK3 activation (necroptosis). RIPK1 can also activate inflammasomes if apoptosis or necroptosis is blocked, yet, *C. rodentium* also blocks this cell death modality via the coordinated action of NleB and EspL.

Although earlier studies have biochemically and functionally elucidated individual *C. rodentium* T3SS effector functions, the interplay between the bacterial effectors and how the host counteracts these bacterial effectors was only incompletely shown. Here, the authors integrate the various aspects into a coherent host: pathogen interplay, demonstrating how pathogens and hosts counteract each others weaponry.

All in all, I think they have addressed all the points of the previous reviewers. I have only a few minor points to add, some of it the authors might want to test experimentally to corroborate their findings and ensure that their cellular system is not flawed due to their use of conditioned medium (that also contains a lot of TNF) of L929 as a source of G-CSF.

Reply 1: We thank the Advisor for their time in reviewing our manuscript, and for appreciating the quality and novelty of our study.

Minor point:

My only suggestion is to streamline the story somewhat making it more accessible to a broader readership.

Reply 2: We agree with the suggestion and have restructured our manuscript for a broader readership.

Of note, the cell line MC38-K is a colorectal carcinoma that expresses good amounts of RIPK3 and readily dies by necroptosis.

Reply 3: Thank you for the suggestion, we will include MC38-K cell in our future studies.

Tone down the conclusion, stating: 'these processes required RIPK1 kinase activity since LDH release or apoptotic caspase activation were significantly reduced in the presence of the RIPK1 kinase inhibitor, Nec-1s'. The authors should be aware that Nec-1s also blocks processes that does not require RIPK1, such as TRIF-mediated or ZBP-induced necroptosis. Nec-1s turns RIPK1 into a dominant-negative molecule through a poorly understood mechanism.

Reply 4: Thank you for highlighting the potential dominant-negative effects of Nec-1s on RIPK1. We have reworded our manuscript accordingly.

Is it possible that TRIF induced IRF3 activation is sufficient to drive Mkl induction? This could easily be tested, or alternatively, the authors may amend their ms to suggest this possibility.

Trif^{-/-} macrophages generally have a fundamentally different baseline level of interferon target genes, such as MLKL. Can the authors evaluate how comparable the baseline levels of Mkl (and other interferon target genes such as ISG15) are between WT, Tnfr1^{-/-} and Trif^{-/-}. If indeed the baseline levels are very different, then it is difficult to conclude that the observed effect is indeed due to TRIF signalling, rather than TNFR1. It may simply reflect the different baseline levels of MLKL. It would be worthwhile to evaluate this.

Reply 5: Thank you for this suggestion. We have considered this possibility and examined the expression of RIPK3 and MLKL in WT and *Trif*^{-/-} macrophages before and after LPS or TNF stimulation and found that the expression of these necroptotic proteins is unaffected by *Trif* deficiency in our system (Manuscript new Fig. EV4B).

Along a similar line, I am surprised to see that their macrophages are more prone to undergo apoptosis. In our hands, macrophages (differentiated with G-CSF) are prone to undergo necroptosis. I have noticed that the authors use conditioned medium from L929 cells as a source of G-CSF. However, the authors should be aware that L929 produce a high amount of TNF in addition to G-CSF. Hence, macrophages that are sensitive to TNF will be killed and only resistant cells will emerge. This may skew their findings.

Reply 6: Thank you for this excellent suggestion. We have now repeated our key experiments with M-CSF-derived macrophages and demonstrate that the results are true for both L929 and M-CSF-derived macrophages. In brief, we confirmed that in M-CSF-derived macrophages, (1) *C. rodentium* triggers apoptosis in a NleE-dependent manner in unprimed macrophages (Rebuttal Fig. 1A), and (2) LPS priming promotes necroptosis and secondary inflammasome activation following $\Delta espL$ infection (Rebuttal Fig. 1B).

Rebuttal Figure 1. *C. rodentium* NleE triggers apoptosis in unprimed macrophages while EspL subverts necroptosis in LPS-primed macrophages. (A-B) BMDM were differentiated with 100 ng/ml recombinant M-CSF for 7 days and challenged with *C. rodentium*, $\Delta nleE$ or $\Delta espL$ at a multiplicity of 20.

(A) Unprimed M-CSF-derived macrophages were infected with *C. rodentium* or $\Delta nleE$ for 5h and precipitated supernatant and cell extracts were combined and analysed by immunoblot. **(B)** M-CSF-derived macrophages were primed with 100ng/ml LPS and infected with *C. rodentium* or $\Delta espL$ for 5h and precipitated supernatant and cell extracts were combined and analysed by immunoblot. Where indicated, macrophages were treated with 100 μ M Nec-1s or 5 μ M GSK'872 30 min before infection.

Arbitrator 2

The manuscript was previously submitted to another journal where it was reviewed and rejected after review. Despite rejection, all three reviewers showed some interest in the study and provided the authors with comments/suggestions. In their point-by-point response, the authors defend the novelty of their findings and describe the editorial changes and new experiments that they are planning to do in the hope for a revision at the EMBO Journal.

While the revision plan of the authors will improve the overall quality of the manuscript, it will not address the concern on the physiological importance of the cell death modalities induced in response to the T3SS effectors for the host immune response against the bacterial infection. As suggested by reviewers 2&3, it seems important to additionally include *in vivo* infection experiments (including mutant *Citrobacter* strains) in RIPK3 KO (or MLKL KO) and RIPK1 KD mice to demonstrate the effect of the effector(s) on the cell death modality, and the importance of its induction for the host immune response.

Reply 7: We thank the Advisor for their time in reviewing our manuscript and our proposed list of experiments. While we agree that the additional *in vivo* experiments comparing bacterial burden in B6 vs *Mkl1*^{-/-} mice following $\Delta espL$ infection can further strengthen our study, we are unable to do so in a timely fashion since it will take several months to import and establish a large enough colony to perform such experiments. However, we will investigate this possibility in a follow-up study.

Additional remarks

Any KO cell line should be studied in comparison to the proper control WT cell line, and not just any WT cells. For instance, *Tnfr1*^{-/-} and *Trif*^{-/-} BMDMs should respectively be compared to *Tnfr*^{+/+} and *Trif*^{+/+} BMDMs isolated from littermates, and not from commercial C57BL/6 mice.

Reply 8: We thank the Advisor for this suggestion. While we agree that littermate controls should be used whenever possible, this approach may not always be practical since we need a number of KO lines in this study. We would also like to highlight that our WT and KO lines are all purchased directly from JAX, which ensures rigorous genotype and genetic control. In addition, we regularly backcross our mice and also compare the response of C57BL/6 to +/+ mice that were generated from heterozygous breeding.

Immunoblot results from different genotypes can only be compared if run on the same gel. This is obviously not the case in Figure 4A.

Reply 9: We absolutely agree with the Advisor that only proteins resolved on the same gel can be used for comparison, and have done so throughout our manuscript. We believe that the Advisor may have misread Figure 4A since the genotypes (B6 vs *Trif*^{-/-} vs *Tnfr1*^{-/-}) are listed on the bottom of the blot.

Line 168: "Host response to counteract NleE subversion of NF- κ B activation". However, NF- κ B inhibition was previously demonstrated not to lead to RIPK1 activation, in contrast to IKK inhibition (see PMID: 37357102). The authors should better define and demonstrate how NleE induces RIPK1 kinase-dependent (and -independent) apoptosis. Blockade of TAB2/3 is not shown. As the authors

make the comparison with *Y. pseudotuberculosis* (which directly target TAK1/IKK), it would be important to evaluate the effect of NleE on TAK1 and IKK.

Reply 10: We agree with the Advisor that it is IKK but not NF- κ B inhibition that leads to RIPK1 activation, thus we have edited our manuscript to improve the clarity of our work. We further agree that demonstrating NleE blockade of TAB2/3-TAK1-IKK signalling will further strengthen our manuscript. However, given the limited sensitivity of TAK1/IKK antibodies, we were unable to probe and assess for TAK1/IKK phosphorylation. Thus, we adopted the methods used in the primary NleE papers (PMID: 20485572, 20126447, 22158122) and examined I κ B α phosphorylation and degradation and expression of NF- κ B target genes such as c-FLIP and MCL-1 in macrophages (**Manuscript new Fig. EV1A-B**) and epithelial cells (**Manuscript new Appendix Fig. S1A-B**). Surprisingly, we observed that NleE blockade of I κ B α phosphorylation and degradation is much weaker than in epithelial cells, however, this is still sufficient to trigger RIPK1-dependent apoptosis in macrophages (**Fig. 1A-B**).

Legend of Figure 4A indicates "Mixed supernatant and lysates were analyzed by immunoblotting". The analysis of the cell lysates and supernatant should not be mixed but analyzed separately.

Reply 11: Upon cell death, numerous cellular proteins including caspases and commonly used loading controls (e.g. tubulin and GAPDH) will be released to the supernatant. Thus, a commonly used method by established labs (PMID: 26375259, 38759620, 38396301) is to precipitate the supernatant and mix them together with the cell lysates to have an overview of substrate processing/PTM, which we favour in our study. This method is also used in several of our previous publications and is accepted by the scientific community (PMID: 38965211, 34260403, 33208362, 30902848).

Dear Kaiwen,

Thank you again for submitting your revised manuscript (EMBOJ-2024-118621R) to The EMBO Journal, and for your patience during re-review. As I have already informed you, it has been sent back to the two advisors/arbitrators who had assessed the previous version of your manuscript, and we have received their comments, which I have already shared with you (they are included again below). I would also like to thank you for the responses to their new comments that you kindly shared with me, and which were very helpful for a balanced and fair decision on the manuscript.

The first advisor is satisfied with the revision and acknowledges that all previous comments have been addressed. The second advisor also recognizes that most of the initially raised concerns have been addressed with the addition of new data, but also points out four outstanding limitations.

Taking into consideration this input, your responses to the comments, the manuscript's review history, and our journal's policies on the process of arbitration, I would like to mention that although we agree with the second advisor that the study would benefit from the remaining points being fully addressed, not all of them will be required to be completely addressed for the publication of the manuscript in The EMBO Journal. In particular:

1. Littermate controls will not be required for the publication of the manuscript.
2. The included data (blots) demonstrating the expression of pro-IL-1b versus cleaved IL-1b are sufficient for publication of the study.
3. We agree that the initially included data in Fig. S1D must be removed from the manuscript, since they could not be robustly repeated during revision. Although additional in vivo experiments will not be required, any over-interpretation of the available data must be avoided, in line with the advisor's recommendation. Toning down the claim/conclusion according to your suggestion ("Our results thus far suggest...") will be sufficient.
4. Similar to the previous point, any speculative conclusions in the absence of conclusive data proving the claims must be avoided. Please make sure that all claims that are not fully supported by the available data must be clearly phrased as speculative, and the limitations/missing data for a conclusive statement to be possible are also mentioned.

I would like to invite you to address these remaining points in a final version of your manuscript as soon as possible. Please include in your resubmission a detailed point-by-point response to the comments describing also any changes to the manuscript in its final version. I would like to remind you that this response letter will be part of the Peer Review File that will be published along with your article.

There are also a few editorial requests/formatting changes that we need from you to address in the final version of your manuscript, before we can proceed with formal acceptance of the manuscript for publication in The EMBO Journal. Please briefly describe how they are addressed in a brief cover letter:

- Please provide a list of up to 5 relevant keywords after the Abstract of your revised manuscript.
- Please change the heading of your "Declaration of interests" statement to "Disclosure and competing interests statement".
- The author contributions statement should be removed from the manuscript file. Instead, we use CRediT to specify the contributions of each author in the journal submission system. Please feel free to use the free text box to provide more detailed descriptions during submission. See also our guide to authors for more information:
<https://www.embopress.org/page/journal/14602075/authorguide#authorshipguidelines>.
- Please provide the details of the authority granting ethics approval of the experiments involving animals, as well as the reference number for approval, in the paragraph "Mice" of your Methods and Protocols.
- Please add the heading "Appendix for" and the manuscript's title on the first page of your Appendix PDF file.
- During our routine pre-acceptance checks, our data editors have raised the following queries regarding figures, data, and legends. Please make sure that all requests below are completely addressed in the final version of your manuscript:
 1. Please note that the legend for Figure 1 is not provided in the sequential manner (the legend for panels E-H is provided before the legend of panels C-D). This needs to be rectified.
 2. Please provide the exact p values in the legends of Figures 1A, C, E; 2A, B; 3B, C, D, G; 4B, C, F; 5A, D; EV1 C, EV2 C-E; EV3 A, B; EV4 A; EV5 A-C.
 3. Please indicate the statistical test used for data analysis in the legends of Figures 1A, C, E; 2A, B; 3C, D, G; 4B, C, F; 5A, D; EV1 C, EV2 C-E; EV3 A, B; EV4 A; EV5 A-C.

4. Please note that information related to "n" is missing in the legend of Figure EV3 A.

- The manuscript section order should be corrected as follows: Title page - Abstract & Keywords - Introduction - Results - Discussion - Methods - Data Availability - Acknowledgements - Disclosure and Competing Interests Statement - References - Figure Legends - main Table(s) (if there are any) - Expanded View Figure Legends.

- Please note that EMBO press papers are accompanied online by:

A) a short (2 sentences) summary of the findings and their significance,

B) 2-5 short bullet points highlighting the key results, and

C) a synopsis image in .jpg or .png format that is exactly 550 pixels wide and 300-600 pixels high (the height is variable). Please note that the text needs to be legible at the final size.

Please upload this information along with your revised manuscript (the text for A and B should be provided in a separate Word file).

Please also note that as part of the EMBO publications' Transparent Editorial Process, The EMBO Journal publishes online a Peer Review File along with each accepted manuscript. This File will be published in conjunction with your paper and will include the referee reports, your point-by-point response and all pertinent correspondence relating to the manuscript. You can opt out of this by letting the editorial office know (contact@embojournal.org). If you do opt out, the Peer Review File link will point to the following statement: "No Peer Review File is available with this article, as the authors have chosen not to make the review process public in this case."

We look forward to seeing a final version of your manuscript as soon as possible. Please let us know if you have any questions and use this link to submit your revision: <https://emboj.msubmit.net/cgi-bin/main.plex>.

Best wishes,

Ioannis

Arbitrator #1:

The authors have addressed all my comments. I am happy with the revisions.

Arbitrator #2:

The authors performed experiments addressing a large part of the concerns raised by the referees of the previous journal. Instead, the authors decided not to consider (or properly address) the comments that I had made, putting me in a difficult position to fully support the revised version of their manuscript.

First, it is good scientific practice to use proper controls in experiments. The fact that the authors do not find it "practical" is certainly not a valid excuse to deviate from the rule. The authors even mention in their response to the Advisor's comments the generation of +/+ mice from heterozygous breeding, meaning that they could have used +/+ and -/- cells isolated from littermates to generate their entire study... While I understand that it is too late now, some key experiments could/should have been confirmed with more appropriate controls. If not using cells from littermates, the authors could have used the CRISPR/Cas9 system to generate KO's from a parental control line.

Second, for proper mechanistic understanding, it appears logical to evaluate separately 1) the cytosolic expression and processing of pro-IL-1b and 2) the subsequent extracellular passive release of active IL-1b (and not its pro-form) when studying the impact of effector proteins (or lack of) on the simultaneous occurrence of different cell death modalities. In the same line of idea, it is important to know if the IL-1b ELISA used by the authors specifically recognize IL-1b, or if it additionally recognizes

pro-IL-1b.

Third, in absence of in vivo experiments comparing bacterial burden in WT vs animals deficient for key cell death effectors (such as MLKL^{-/-} or RIPK1 KD), the authors cannot make claims on the role of cell death as a host protective mechanism. Still, they write in line 198: "Our results thus far demonstrated RIPK1-dependent caspase-8 activation as a conserved host protective mechanism upon pathogen blockade of innate immune signalling." Results from Fig. S1D (which do not seem to be called in the manuscript) also do not show higher CFU value upon infection with the nleE-lacking strain.

Fourth, the authors assume that nleE triggers RIPK1 activation through inhibition of IKK-mediated phosphorylation-dependent repression of RIPK1, but they do not provide any data demonstrating that it is the case. They just show that nleE has a minor effect on IκBα degradation. What would be the cytotoxic effect of nleE in macrophages pretreated with IKK inhibitor?

Dear Editors and Advisors,

We thank you for your time and the constructive feedback to improve our manuscript.

Point-by-point reply to Advisor comments

Arbitrator 1

The authors have addressed all my comments. I am happy with the revisions.

Reply 1: We thank Advisor 1 for all the suggestions and feedback during the peer review. Your suggestions have strengthened our manuscript.

Arbitrator 2

The authors performed experiments addressing a large part of the concerns raised by the referees of the previous journal. Instead, the authors decided not to consider (or properly address) the comments that I had made, putting me in a difficult position to fully support the revised version of their manuscript.

First, it is good scientific practice to use proper controls in experiments. The fact that the authors do not find it "practical" is certainly not a valid excuse to deviate from the rule. The authors even mention in their response to the Advisor's comments the generation of $+/+$ mice from heterozygous breeding, meaning that they could have used $+/+$ and $-/-$ cells isolated from littermates to generate their entire study... While I understand that it is too late now, some key experiments could/should have been confirmed with more appropriate controls. If not using cells from littermates, the authors could have used the CRISPR/Cas9 system to generate KOs from a parental control line.

Reply 2: We agree with the Advisor that it is good scientific practice to use proper controls for our experiments. Our approach of using inbred C57BL/6 mice as a source of control cells is practised by multiple laboratories globally, including the two landmark papers that described the discovery of GSDMD (PMID: 26375259, 26375003) and many others that were published in EMBOJ (PMID: 39300211, 35112724, 36647737). In addition, majority of our transgenic animals were purchased directly from JAX, which have been through rigorous genetic quality control and genotyping. Lastly, we have also used chemical inhibitors on our C57BL/6 cells (GSK'872 on RIPK3 and MCC950 on NLRP3) to validate our findings. Thus, we are confident about the results from our study.

Second, for proper mechanistic understanding, it appears logical to evaluate separately 1) the cytosolic expression and processing of pro-IL-1b and 2) the subsequent extracellular passive release of active IL-1b (and not its pro-form) when studying the impact of effector proteins (or lack of) on the simultaneous occurrence of different cell death modalities. In the same line of idea, it is important to know if the IL-1b ELISA used by the authors specifically recognize IL-1b, or if it additionally recognizes pro-IL-1b.

Reply 3: We agree with the Advisor on this. The IL-1b ELISA that we used in our study does not discriminate the pro and cleaved forms of IL-1b, thus, we have specifically demonstrated the expression of pro-IL-1b versus cleaved IL-1b in several western blots (Fig. 4A, 5E). We believe that the Advisor might have missed these data.

Third, in absence of in vivo experiments comparing bacterial burden in WT vs animals deficient for key cell death effectors (such as MLKL $-/-$ or RIPK1 KD), the authors cannot make claims on the role of cell

death as a host protective mechanism. Still, they write in line 198: "Our results thus far demonstrated RIPK1-dependent caspase-8 activation as a conserved host protective mechanism upon pathogen blockade of innate immune signalling." Results from Fig. S1D (which do not seem to be called in the manuscript) also do not show higher CFU value upon infection with the *nleE*-lacking strain.

Reply 3: The purpose of Fig. S1D in the initial submission was to demonstrate that NleE promotes caspase-8/3 activation *in vivo* (Fig 1H; first submission) even at a time point where bacterial load between WT and $\Delta nleE$ remains comparable (Fig. S1D; first submission). However, since caspase-8/3 activation was not as robust in subsequent repeat experiments during revision, we have decided to remove this data. We agree that comparing bacterial load in WT vs *Mlkl*^{-/-} animals will further strengthen our data, and we acknowledge that this is a limitation of our study. We aim to investigate this more thoroughly using conditional knockout animals in our subsequent study. Thus, we have amended the text in our revised manuscript to refrain from speculative conclusion. Specifically, line 202 which now states 'Our results thus far *suggest* RIPK1-dependent caspase-8 activation as a conserved host protective mechanism upon pathogen blockade of innate immune signalling.'

Fourth, the authors assume that *nleE* triggers RIPK1 activation through inhibition of IKK-mediated phosphorylation-dependent repression of RIPK1, but they do not provide any data demonstrating that it is the case. They just show that *nleE* has a minor effect on I κ B α degradation. What would be the cytotoxic effect of *nleE* in macrophages pretreated with IKK inhibitor?

Reply 4: We were also surprised that NleE has a minor role in suppressing I κ B α degradation, which is in contrast to the well-established functions of YopJ, a bacterial acetyltransferase from *Y. pseudotuberculosis*. Despite this, loss of NleE restores cellular viability and abrogates caspase-8 activation upon macrophage and epithelial cell infection, thus, we are confident that detection of NleE indeed triggers the RIPK1/caspase-8 signalling axis. We and others previously demonstrated that macrophages treated with IKKi and LPS will activate the RIPK1/caspase-8 signalling axis (PMID: 30361383, 30381458, 33208362, 36647737), thus, we anticipate that treating cells with a IKK inhibitor and NleE mutant (which will provide LPS) will similarly trigger cell death.

Dear Kaiwen,

Congratulations on an excellent work! I am very pleased to inform you that your manuscript has now been accepted for publication in The EMBO Journal. Thank you for comprehensively addressing the initially raised concerns and all editorial requests.

If you have any questions, please do not hesitate to contact the Editorial Office. Thank you for your contribution to The EMBO Journal. Working with you has been a pleasure!

Best wishes,

Ioannis
